# EXPLANATIONS OF GNN ON EVOLVING GRAPHS VIA AXIOMATIC LAYER EDGES

**Yazheng Liu, Sihong Xie**[*]
The Hong Kong University of Science and Technology (Guangzhou), Guangzhou, China
`yliu533@connect.hkust-gz.edu.cn, sihongxie@hkust-gz.edu.cn`

## ABSTRACT

Graphs are ubiquitous in social networks, chemical molecules, and financial data, where Graph Neural Networks (GNNs) achieve superior predictive accuracy. Graphs can be evolving, while understanding how GNN predictions respond to the evolution provides significant insight and trust. We explore the problem of explaining evolving GNN predictions due to continuously changing edge weights. We introduce a layer edge-based explanation to balance explanation fidelity and interpretability. We propose a novel framework to address the challenges of axiomatic attribution and the entanglement of multiple computational graph paths due to continuous change of edge weights. We first design an axiomatic attribution of the evolution of the model prediction to message flows, then develop Shapley value to fairly map message flow contributions to layer edges. We formulate a novel optimization problem to find the critical layer edges based on KL-divergence minimization. Extensive experiments on eight datasets for node classification, link prediction, and graph classification tasks with evolving graphs demonstrate the better fidelity and interpretability of the proposed method over the baseline methods. The code is available at https://github.com/yazhengliu/Axiomatic-Layer-Edges/tree/main.

## 1 INTRODUCTION

Graph neural networks (GNNs) achieve superior performance in many graph learning tasks, such as social network modeling (Kipf & Welling, 2017), molecule property prediction (Wu et al., 2018), knowledge graph embedding (Wang et al., 2019a), fraud detection (Wang et al., 2019b), and recommendation systems (Ying et al., 2018). Due to the complex message calculation, aggregation, and nonlinear update mechanisms of GNN, they are usually deep, highly nonlinear, and complex. It is desirable to make GNN predictions transparent to humans (Ying et al., 2019; Schnake et al., 2020). For example, a user may want to know why a recommendation is made by GNNs to ensure no breach of sensitive information (e.g., age and gender) (Li et al., 2021); a GNN-based rumor or spam detector should explain why an user account is suspicious (Lai & Tan, 2019).

In the real world, graphs are usually evolving, with input edge weight continuously changing (including addition and deletion of edges/nodes), leading to changes in GNNs model predictions, see Figure 1. For example, in rumor detection task, as new tweets or product reviews are posted over time, the edge weights are also continuous changing due to some factors such as rumor dissemination speed and user interaction frequency. Consequently, the suspiciousness of an account changes accordingly. Let $G_0 \to G_1$ be *any* two snapshots where the source graph $G_0$ evolves to the destination graph $G_1$ with the edges weights changed continuously. Accordingly, $\Pr(Y|G_0; \boldsymbol{\theta})$ will evolve to $\Pr(Y|G_1; \boldsymbol{\theta})$, and we aim to attribute the change in

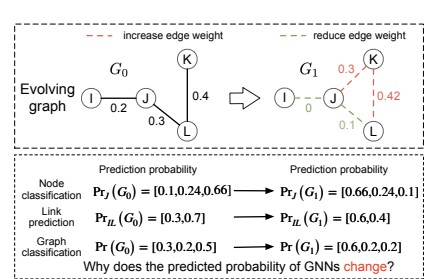

Figure 1: *Top*: The weights of the input edges in the evolution graph change continuously. *Bottom*: Altered weights leads to the changes in GNNs predictions for node classification, link prediction, and graph classification tasks.

---

[*]Sihong Xie is the the corresponding author

$\Pr(Y|G; \boldsymbol{\theta})$ to elements changed (such as input edges) in $G_0 \rightarrow G_1$. With this tool, decision-makers can understand this evolution. For example, what specific rumor spread pattern changes lead to prediction shift.

Explanations of GNN models on evolving graphs must account for both *Interpretability* (Ras et al., 2018) and *Fidelity* (Yuan et al., 2020b). *Interpretability* ensures the selected key elements are easy for users to understand, while *Fidelity* ensures these elements faithfully reflect shifts in model predictions. As GNNs capture complex relationships through multiple message passing and aggregation steps, the explanation should also reflect these complex interaction between nodes and edges, especially when the edge weights are changing. However, such complex evolving interactions involve comparing many node features and edges on the initial and destination graphs, making the explanations complicated for users to understand. Thus, there is often a trade-off between these two factors (see Figure 2 (a)). Various GNN explanation methods have been proposed, including GN-NExplainer (Ying et al., 2019), PGExplainer (Luo et al., 2020), and FlowX (Gui et al., 2023) to select important input edges, layer edges, or message flows, respectively. We consider three types of explanations: message flow, input edges and layer edges. For the **message flow explanations**, if the GNN model with $T$ layers, the message flows with $T + 1$ nodes contain the precise computational process of GNN predictions, leading to the highest fidelity (Gui et al., 2023). However, understanding message flows requires users to be familiar with the multi-layers information ag-

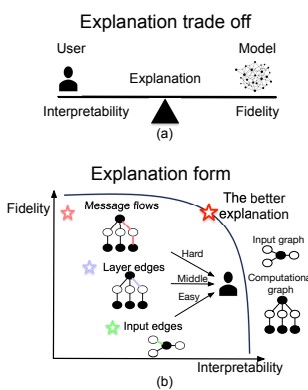

Figure 2: (a) Explanations should have high *Fidelity* and *Interpretability*, accurately representing the model while remaining user-friendly. However, there is often a trade off. (b) The *Fidelity* and *Interpretability* of different explanation forms.

gregation and transformation, which can lead to cognitive overload (Anderson et al., 2020) for those users unfamiliar with GNNs, even for the GNNs model designer, especially when node degrees are high, resulting in the worst *Interpretability*. For the **input edges explanations**, the input edges are directly related to the specific graph elements, often corresponding to real-word concepts, which users find easier to understand, offering the highest *Interpretability*. However, as input edges contain fewer computational process of GNNs predictions, they offer the worst *Fidelity*. For the **layer edge explanations**, the layer edges capture aggregated message information used in the GNN's calculations compared to the input edges, leading to the higher Fidelity than input edges. Although the layer edges transmit information that has been processed through multiple layers, it is easy to understand compared with the message flow explanation, resulting in the higher *Interpretability* than the message flows. Figure 2(b) illustrates the *Fidelity* and *Interpretability* of different explanation forms.

For dynamic graphs, explaining the change from $\Pr(Y|G_0; \boldsymbol{\theta})$ to $\Pr(Y|G_1; \boldsymbol{\theta})$ has several challenges: 1) To explain the changes in predicted probability, it is necessary to understand shifts in *logits* (the final GNN layer output before activation) between the $G_0$ and $G_1$. These *logits* changes can be derived mathematically and mapped to probability shifts. However, existing methods focus on static graphs. Although the static graphs can be considered as an evolution from an empty graph (e.g. $G_1$ evolving from the $G_{\text{empty}}$), these methods only explain the changes in *logits* between $G_{\text{empty}}$ and $G_1$. They ignore the differences between $G_{\text{empty}}$ and $G_0$, leading to inaccurate contributions that fail to explain the evolution of prediction probability, as shown in Figure 3(a). 2) To balance *Interpretability* and *Fidelity*, we provide the layer edges as explanation, requiring a mapping function to convert the message flows contributions into layer edges. Existing methods overlook the fact that layer edges in message flows may contribute differently. The contribution of a layer edge is influenced not only by its associated weight but also by the hidden vector of the node connected to it. See the example in Figure 3(b). Fairly attributing the contributions of message flows to the layer edges is also the key challenge. 3) To ensure the explanations should be understandable to humans, it is important to select a small number of layer edges. The layer edges selected by the top-$K$ in the existing methods may not faithfully represent the model's behavior, as demonstrated in the case shown in Figure 3(c). Selecting the small number of layer edges that provide a faithful explanation of the model is also the challenge.

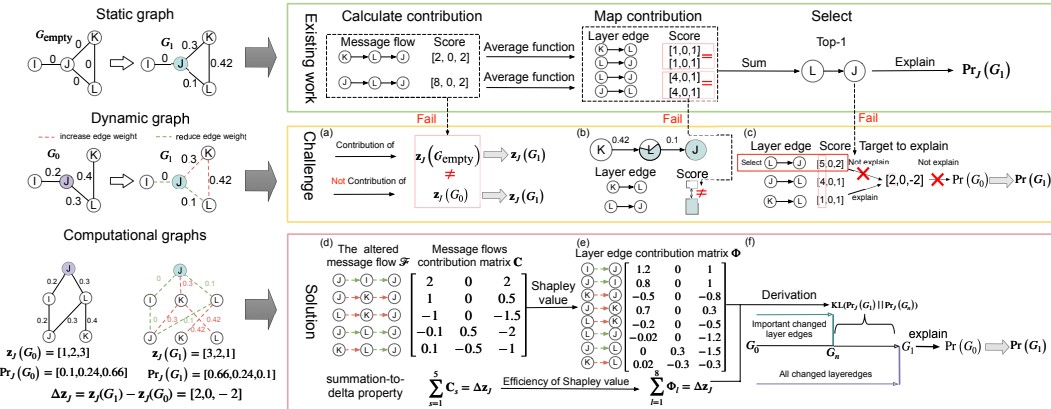

Figure 3: **Top**: The steps for selecting important layer edges as explanations on static graphs in the existing methods. **Middle**: The challenges of using the existing methods to explain the evolution of prediction probability. (a) The existing methods can only explain the changes in *logits* between $G_{empty}$ and $G_1$, but cannot explain the changes in *logits* between $G_1$ and $G_0$. (b) Layer edges may contribute differently not equally. (c) The top-$K$ selection cannot faithfully explain the prediction. **Bottom**: The proposed method to explain the evolution of prediction probability. (d) We calculate the contribution of message flows, ensuring the summation-to-delta property. (e) We use Shapley value to attribute the contribution of message flows to layer edges, preserving the summation-to-delta property. (f) Leveraging the summation-to-delta property, we derive KL divergence and formulate an optimization problem to faithfully select important layer edges as explanations.

To address these challenges: 1) We formula the changes in hidden vectors on $G_0$ and $G_1$ and apply the multipliers and chain rule of DeepLIFT to assign the changes in *logits* to message flows, ensuring that their contributions follow the summation-to-delta property. 2) We frame the mapping of message flows to layer edges as an allocation problem. Then, we apply the Shapley value to fairly attribute contributions, ensuring that layer edges' contributions also satisfy the summation-to-delta property due to the Shaplye value's efficiency. 3) Based on summation-to-delta property, we derive the Kullback-Leibler divergence, and define an objective function to map changes in *logits* to shifts in predicted probability. By solving this optimization problem, we select a small number of layer edges that faithfully explain the evolution of predicted probability. Extensive experiments on eight datasets for node classification, link prediction and graph classification tasks with evolving graphs show the effectiveness of our method in explaining the evolution of the predicted probability. Our methods empirically outperforms five popular, state-of-the-art baselines across the three graph tasks.

## 2 PRELIMINARIES

### 2.1 GRAPH NEURAL NETWORKS

For *node classification*, consider a trained GNN with $T$ layers that predicts the class distribution of each node $J \in \mathcal{V}$ in a graph $G = (\mathcal{V}, \mathcal{E})$. Let $e_{IJ}$ denote a directed edge from node $I$ to node $J$. Let $A$ represent the adjacency matrix of graph $G$. The element $a_{IJ}$ of $A$ represents the weight of the edge $e_{IJ}$, and $a_{IJ} = 0$ indicates that $e_{IJ}$ does not exist. Let $\mathcal{N}(J)$ denote the neighbors of node $J$. At layer $t$ ($t = 1, \ldots, T$), for node $J$, the GNN computes hidden vector $\mathbf{h}_J^t$ using messages received from its neighbors:

$$\mathbf{z}_J^t = f_{\text{UPDATE}}^t(f_{\text{AGG}}^t(\{\mathbf{h}_J^{t-1}, \mathbf{h}_I^{t-1} : I \in \mathcal{N}(J)\}), \boldsymbol{\theta}^t), \tag{1}$$
$$\mathbf{h}_J^t = \text{NonLinear}(\mathbf{z}_J^t), \tag{2}$$

where $f_{\text{AGG}}^t$ aggregates the messages from all neighbors, using element-wise operations, such as sum, average, or maximum. The function $f_{\text{UPDATE}}^t$ maps the aggregated messages $f_{\text{AGG}}^t$ to $\mathbf{z}_J^t$ with parameters $\boldsymbol{\theta}^t$. For layer $t \in \{1, \ldots, T-1\}$, ReLU is used as the nonlinear activation function. At input layer, $\mathbf{h}_J^0$ is the node feature vector $\mathbf{x}_J$. At layer $T$, the logits are given by $\mathbf{z}_J^T \triangleq \mathbf{z}_J(G)$. The logits $\mathbf{z}_J(G)$ is mapped to the class distribution $\Pr(Y_J | G; \boldsymbol{\theta})$ using the softmax or sigmoid function.

For *link prediction*, we concatenate $\mathbf{z}_I^T$ and $\mathbf{z}_J^T$ as input to a linear layer to compute the logits:

$$\mathbf{z}_{IJ} = \left\langle \left[ \mathbf{z}_I^T ; \mathbf{z}_J^T \right], \boldsymbol{\theta}_{LP} \right\rangle. \tag{3}$$

Since link prediction is a binary classification problem, $\mathbf{z}_{IJ}$ is mapped to the probability of edge $(I, J)$ existence using the sigmoid function.

For *graph classification*, the average pooling of $\mathbf{z}_J(G)$ across all nodes in graph $G$ produces a single vector representation $\mathbf{z}(G)$ for classification.

## 2.2 THE MESSAGE FLOW VIEW OF GNN

**Layer Edges**: Given the adjacency matrix $A^t$ at layer $t$, the layer edge $a_{IJ}^t$ in this matrix represents the message carrier with which the message passes from node $I$ to node $J$. The set of layer edges, is defined as $\boldsymbol{\mathcal{A}} = \{ \cdots, a_{UV}^1, \cdots, a_{UV}^t, \cdots, a_{UV}^T, \cdots \}$ and $|\boldsymbol{\mathcal{A}}| = |\mathcal{E}| \times T$. For example, assuming the nodes in $G_0$ in Figure 1 have no self-connections, and $T = 2$, $\boldsymbol{\mathcal{A}} = \{ a_{IJ}^1, a_{IJ}^2, a_{JI}^1, a_{JI}^2, a_{JL}^1, a_{JL}^2, a_{LJ}^1, a_{LJ}^2, a_{KL}^1, a_{KL}^2, a_{LK}^1, a_{LK}^2 \}$.

**Message Flow**: In a $T$-layer GNN model, let $\mathcal{F} = (I, M, \ldots, U, V, \ldots, L, J)$ denote the message flow starting from node $I$ in the input layer, and sequentially passing messages through node $M, \ldots, U, V, \ldots, L$, until reaching node $J$ in the final layer $T$. The corresponding layer edges can be represented as $(a_{IM}^1, \ldots, a_{UV}^t, \ldots, a_{LJ}^T)$. Let $\mathcal{F}[t]$ denote the node at the layer $t$ in this message flow $\mathcal{F}$, where $t = 0, \ldots, T$, $t = 0$ denotes the input layer. For example, for the message flow $\mathcal{F} = (I, L, J)$, the corresponding layer edges are $(a_{IL}^1, a_{LJ}^2)$, $\mathcal{F}[0] = I, \mathcal{F}[1] = L, \mathcal{F}[2] = J$.

Table 1: Symbols and their meanings

## 2.3 EVOLVING GRAPHS

Let $\tau \in \{0, 1\}$ denote the time steps of two graph snapshots, where $G_\tau = (\mathcal{V}^\tau, \mathcal{E}^\tau)$ represents the graph. The adjacency matrix of $G_\tau$ is denoted as $A^\tau$, with element $a_{UV}^\tau$ representing the weight of input edge. Additionally, $A^{\tau,t}$ refers to the adjacency matrix at layer $t$, with element $a_{UV}^{\tau,t}$ representing the weight of layer edge. Let $\mathbf{h}_J^{\tau,t}$ and $\mathbf{z}_J^{\tau,t}$ denote the hidden vector and relevant vector of node $J$ at layer $t$ in graphs $G_\tau$. As the graph evolves from $G_0$ to $G_1$, we define

| Symbols | Definitions and Descriptions |
|---|---|
| $\tau$ | $\tau \in \{0, 1\}$ indicate the time steps |
| $a_{UV}^\tau$ | The weight of input edge in graph $G_\tau$ |
| $a_{UV}^{\tau,t}$ | The weight of layer edge in graph $G_\tau$ at layer $t$ |
| $\Delta\boldsymbol{\mathcal{F}}$ | The set of altered message flows |
| $\Delta\boldsymbol{\mathcal{A}}$ | The set of altered layer edges |
| $\mathbf{h}_J^{\tau,t}$ | The hidden vector of node $J$ at layer $t$ in $G_\tau$ |
| $\mathbf{z}_J^{\tau,t}$ | The relevant vector of node $J$ at layer $t$ in $G_\tau$ |
| $\Delta a_{UV}^t$ | The difference in weight of layer edge between $G_0$ and $G_1$ |
| $\Delta\mathbf{z}_J^t$ | The difference in relevant vector $\mathbf{z}_J^t$ between $G_0$ and $G_1$ |
| $\Delta\mathbf{h}_J^t$ | The difference in hidden vector $\mathbf{h}_J^t$ between $G_0$ and $G_1$ |
| $\mathbf{C}$ | Contribution of the message flows to $\Delta\mathbf{z}$ |
| $\Phi$ | Contribution of the layer edges to $\Delta\mathbf{z}$ |

the change in layer edge weight as $\Delta a_{UV}^t = a_{UV}^{1,t} - a_{UV}^{0,t}$. Similarly, Let $\Delta\mathbf{h}_J^t = \mathbf{h}_J^{1,t} - \mathbf{h}_J^{0,t}$ and $\Delta\mathbf{z}_J^t = \mathbf{z}_J^{1,t} - \mathbf{z}_J^{0,t}$ denote the difference in the hidden vectors and relevant vectors at layer $t$ between graph $G_0$ and $G_1$. Let $\Delta\mathbf{z}_J \triangleq \Delta\mathbf{z}_J^T$ denote the difference in logits of node $J$ between $G_0$ and $G_1$.

We assume that the edge weights change continuously, including edge additions and removals. Let $\Delta\mathcal{E}$ be the set of altered edges: $\Delta\mathcal{E} = \{ e_{UV} : a_{UV}^0 \neq a_{UV}^1, U, V \in \mathcal{V} \}$. Let $\Delta\boldsymbol{\mathcal{A}}$ represent the set of altered layer edges: $\Delta\boldsymbol{\mathcal{A}} = \{ a_{UV}^t : a_{UV}^{0,t} \neq a_{UV}^{1,t}, t \in \{1, \ldots, T\}, U, V \in \mathcal{V} \}$. If the weight of a layer edge within a message flow changes between $G_0$ and $G_1$, the message flow is also altered. Let $\Delta\boldsymbol{\mathcal{F}}$ denote the set of altered message flows: $\Delta\boldsymbol{\mathcal{F}} = \{ \mathcal{F} : \mathcal{F} = (\mathcal{F}[0], \ldots, \mathcal{F}[t] \ldots \mathcal{F}[T]), a_{\mathcal{F}[t-1]\mathcal{F}[t]}^{0,t} \neq a_{\mathcal{F}[t-1]\mathcal{F}[t]}^{1,t}, t = 1, \ldots, T \}$. As $G_0 \to G_1$, the prediction evolves from $\Pr(Y|G_0)$ to $\Pr(Y|G_1)$. The set $\Delta\boldsymbol{\mathcal{F}}$ causes this evolution, as the information propagated by these massage flows differs between the source and destination graphs. Thus, $\Delta\boldsymbol{\mathcal{F}}$ provides an explanation of the evolution with 100 % Fidelity, without any loss of information. However, due to lack of Interpretability, it is difficult for humans to understand. The complexity of $\Delta\boldsymbol{\mathcal{F}}$ can increase significantly due to changes in edge weights connected to high degree nodes. Small perturbations in graph can cause $\Delta\boldsymbol{\mathcal{A}}$ to become large, further impacting the complexity of $\Delta\boldsymbol{\mathcal{F}}$. As a result, $\Delta\boldsymbol{\mathcal{F}}$ can be not serve as a good explanation.

## 3 METHOD

We propose a method to study the explainability of the evolution from $\Pr(Y|G_0; \boldsymbol{\theta})$ to $\Pr(Y|G_1; \boldsymbol{\theta})$. To address the challenge in Figure 3, we first derive the $\Delta h_V^t, t = 1, \ldots, T$. We then apply the multipliers and chain rule from DeepLIFT to calculate the contributions of altered message flows, and obtain the contribution matrix $\mathbf{C}$. This contribution matrix satisfy the summation-to-delta property, i.e. $\Delta \mathbf{z}_J = \sum_{s=1}^{|\Delta \mathcal{F}|} \mathbf{C}_s$. However, due to the non-linearity of softmax function and KL divergence, $\Delta \mathbf{z}_J$ can not linearly represent the evolution from $\Pr(Y|G_0; \boldsymbol{\theta})$ to $\Pr(Y|G_1; \boldsymbol{\theta})$. To resolve this, We map the $\Delta \mathbf{z}_J$ to the evolution from $\Pr(Y|G_0; \boldsymbol{\theta})$ to $\Pr(Y|G_1; \boldsymbol{\theta})$ through mathematical derivation, utilizing the summation-to-delta property, as illustrated in Figure 3(d). Next, to fairly attribute the contribution of message flows to layer edges, we employ the Shapley values and compute the contribution of layer edges, denoted as $\Phi$. Due to the efficiency of Shapley values, $\Phi$ also holds the summation-to-delta property, i.e. $\Delta \mathbf{z}_J = \sum_{l=1}^{|\Delta \mathcal{A}|} \Phi_l$, as shown in Figure 3(e). Finally, to faithfully select the important layer edges, we derive the KL divergence and design the optimization problem, leveraging the summation-to-delta property, as shown in Figure 3(f).

### 3.1 CALCULATE THE CONTRIBUTION OF MESSAGE FLOWS

To demonstrate the calculation of contribution values, we focus on the node classification task. Details for calculating contribution values in link prediction and graph classification tasks can be found in Appendix A.2.2 and A.2.3, respectively. We use DeepLIFT (Shrikumar et al., 2017) to calculate the contribution values of message flows and ensure the $\Delta \mathbf{z}_J = \sum_{s=1}^{|\Delta \mathcal{F}|} \mathbf{C}_s$, which existing work did not do with continuously changing edge weights.

#### 3.1.1 DEEPLIFT

We introduce the multipliers and chain rules from DeepLIFT (Shrikumar et al., 2017) which are used to calculate the contributions of message flows. While DeepLIFT is designed to evaluate contributions at the neuron level in multi-layer preceptron models, we extend it to a vectorized representation for computational efficiency. Let $\tau \in \{0, 1\}$ denote the time step. Let $\mathbf{h}^{\tau,t} \in \mathbb{R}^{1 \times n}$ and $\mathbf{h}^{\tau,t+1} \in \mathbb{R}^{1 \times m}$ represent the hidden layer vector at layer $t$ and $t+1$, at time step $\tau$, respectively. The vector $\mathbf{h}^{\tau,t+1}$ is computed as $\mathbf{h}^{\tau,t+1} = f(\mathbf{h}^{\tau,t})$, where $f(\mathbf{h}^{\tau,t}) = \mathbf{h}^{\tau,t} \boldsymbol{\theta}^t$ for a linear function, with $\boldsymbol{\theta}^t \in \mathbb{R}^{n \times m}$ as the weight matrix, otherwise, $f$ is the nonlinear activation function. The *difference-from-reference* is defined as $\Delta \mathbf{h}^{t+1} = \mathbf{h}^{1,t+1} - \mathbf{h}^{0,t+1}$ and $\Delta \mathbf{h}^t = \mathbf{h}^{1,t} - \mathbf{h}^{0,t}$. DeepLIFT defines multiplier as follows:

$$\mathbf{m}_{\Delta \mathbf{h}^t \Delta \mathbf{h}^{t+1}} = \begin{cases} \boldsymbol{\theta}^t \in \mathbb{R}^{n \times m} & \text{linear layer} \\ \Delta \mathbf{h}^{t+1} / \Delta \mathbf{h}^t \in \mathbb{R}^{1 \times n} & \text{nonlinear activation} \end{cases} \quad (4)$$

$/$ denotes element-wise division. The following relationship holds: $\Delta \mathbf{h}^t \times \mathbf{m}_{\Delta \mathbf{h}^t \Delta \mathbf{h}^{t+1}} = \Delta \mathbf{h}^{t+1}$, where $\times$ represents matrix multiplication if $f$ is linear, and element-wise multiplication if $f$ is nonlinear. DeepLIFT defines the chain rules as

$$\begin{aligned} \Delta \mathbf{h}^T &= \Delta \mathbf{h}^{T-1} \mathbf{m}_{\Delta \mathbf{h}^{T-1} \Delta \mathbf{h}^T} = \Delta \mathbf{h}^{T-2} \mathbf{m}_{\Delta \mathbf{h}^{T-2} \Delta \mathbf{h}^{T-1}} \mathbf{m}_{\Delta \mathbf{h}^{T-1} \Delta \mathbf{h}^T} \\ &= \Delta \mathbf{h}^0 \mathbf{m}_{\Delta \mathbf{h}^0 \Delta \mathbf{h}^1} \ldots \mathbf{m}_{\Delta \mathbf{h}^{T-1} \Delta \mathbf{h}^T}. \end{aligned} \quad (5)$$

According to chain rule and multipliers, we can calculate the contribution of message flows.

#### 3.1.2 DEEPLIFT FOR GNN

DeepLIFT has been applied to GNNs with the addition and removal of edges (Liu et al., 2024). However, existing methods assume discrete graph evolution, leading to incorrect contribution calculations for message flows in continuously evolving GNNs. This is because, *difference-from-reference*, used in the calculation process of DeepLIFT, becomes more complex and inconsistent in continuously evolving GNNs. To address this challenge, for a given message flow $\mathcal{F} \in \Delta \mathcal{F}$, we derive $\Delta \mathbf{h}_{\mathcal{F}[t]}^t, t = 0, \cdots, T$ based on the propagation rules of GNNs. Then, we use multipliers and the chain rule defined by DeepLIFT to calculate the contribution value of message flow.

In GNNs, $\mathbf{z}_V^t = \sum_{U \in \mathcal{N}(V)} a_{UV}^t \mathbf{h}_U^{t-1} \boldsymbol{\theta}^t$, where $\mathbf{z}_V^t$ depends to two factors: the information from the neighboring node $\mathbf{h}_U^{t-1}$ and the edge weight $a_{UV}^t$. Consequently, $\Delta \mathbf{z}_V^t$ is influenced by changes

in both $\Delta\mathbf{h}_U^{t-1}$ and $\Delta a_{UV}^t$. Specifically, $\Delta\mathbf{h}_U^{t-1}$ propagates to node $V$, altering $V$'s information. Additionally, $\Delta a_{UV}^t$ affect the extent of information aggregation from node $U$ to node $V$. The formula for $\Delta\mathbf{z}_V^t$ is given by:

$$
\sum_{U\in\mathcal{N}(V)}\left(a_{UV}^{1,t}\mathbf{h}_U^{1,t-1}\boldsymbol{\theta}^t - a_{UV}^{0,t}\mathbf{h}_U^{0,t-1}\boldsymbol{\theta}^t\right) = \sum_{U\in\mathcal{N}(V)}\left((a_{UV}^{0,t}+\Delta a_{UV}^t)\mathbf{h}_U^{1,t-1}\boldsymbol{\theta}^t - a_{UV}^{0,t}\mathbf{h}_U^{0,t-1}\boldsymbol{\theta}^t\right)
$$
$$
= \sum_{U\in\mathcal{N}(V)}a_{UV}^{0,t}\left(\mathbf{h}_U^{1,t-1}-\mathbf{h}_U^{0,t-1}\right)\boldsymbol{\theta}^t + \Delta a_{UV}^t\mathbf{h}_U^{1,t-1}\boldsymbol{\theta}^t = \sum_{U\in\mathcal{N}(V)}a_{UV}^{0,t}\Delta\mathbf{h}_U^{t-1}\boldsymbol{\theta}^t + \Delta a_{UV}^t\mathbf{h}_U^{1,t-1}\boldsymbol{\theta}^t.
$$
(6)

Eq. (6) explains the cause of $\Delta\mathbf{z}_V^t$ by decomposing the change into two parts. The first term $a_{UV}^{0,t}\Delta\mathbf{h}_U^{t-1}\boldsymbol{\theta}^t$ represents the propagation of $\Delta\mathbf{h}_U^{t-1}$ from node $U$ to node $V$ when the the layer edge weight $a_{UV}^t$ remains unchanged. Since $\mathbf{h}_U^{1,t-1} = \mathbf{h}_U^{0,t-1} + \Delta\mathbf{h}_U^{t-1}$, the second term can be rewritten as $\Delta a_{UV}^t\left(\mathbf{h}_U^{0,t-1}+\Delta\mathbf{h}_U^{t-1}\right)\boldsymbol{\theta}^t$, showing that the change in edge weight transfers both $\mathbf{h}_U^{0,t-1}$ and $\Delta\mathbf{h}_U^{t-1}$ to node $V$, contributing to $\Delta\mathbf{z}_V^t$.

According to Eq. (6), and by applying the chain rule and the multipliers defined in Eq. (4), the contribution of a message flow can be computed through layer by layer decomposition. For a given flow $\mathcal{F}$ in $\Delta\mathcal{F}$, $\mathcal{F}[t]$ denotes the node at $t$ layer in $\mathcal{F}$. The formula for calculating the contribution of this message flow is as follows (detailed derivations and examples are provided in Appendix A.2.1):

$$
\mathbf{C}_s = \sum_{t=0}^{T-1}\left(a_{\mathcal{F}[0]\mathcal{F}[1]}^{1,1}a_{\mathcal{F}[1]\mathcal{F}[2]}^{1,2}\cdots\Delta a_{\mathcal{F}[t]\mathcal{F}[t+1]}^{t+1}a_{\mathcal{F}[t+1],\mathcal{F}[t+2]}^{0,t+2}\cdots a_{\mathcal{F}[T-1],\mathcal{F}[T]}^{0,T}\right.
$$
$$
\left.\mathbf{h}_{\mathcal{F}[0]}^{1,0}\frac{\mathbf{h}_{\mathcal{F}[1]}^{1,1}}{\mathbf{z}_{\mathcal{F}[1]}^{1,1}}\cdots\frac{\mathbf{h}_{\mathcal{F}[t]}^{1,t}}{\mathbf{z}_{\mathcal{F}[t]}^{1,t}}\boldsymbol{\theta}^t\frac{\Delta\mathbf{h}_{\mathcal{F}[t+1]}^{t+1}}{\Delta\mathbf{z}_{\mathcal{F}[t+1]}^{t+1}}\boldsymbol{\theta}^{t+1}\cdots\frac{\Delta\mathbf{h}_{\mathcal{F}[T-1]}^{T-1}}{\Delta\mathbf{z}_{\mathcal{F}[T-1]}^{T-1}}\boldsymbol{\theta}^T\right),
$$
(7)

where ratios denote the element-wise division, and $\mathbf{C}\in\mathbb{R}^{|\Delta\mathcal{F}|\times c}$ denotes the the contribution matrix of message flows. Let $s$ denote $s$-th flow $\mathcal{F}$ in $\Delta\mathcal{F}$ to $\Delta\mathbf{z}_J$. Due to the multipliers and chain rules, the contribution matrix $\mathbf{C}$ satisfies the summation-to-delta property, i.e. $\sum_{s=1}^{|\Delta\mathcal{F}|}\mathbf{C}_s = \Delta\mathbf{z}_J$. Leveraging this property, we can map the $\Delta\mathbf{z}_J$ to the evolution from $\Pr(Y|G_0;\boldsymbol{\theta})$ to $\Pr(Y|G_1;\boldsymbol{\theta})$.

## 3.2 Apply the Shapley value to map message flow contributions to layer edges

Message flows are difficult for humans to understand and challenging to evaluate. In GNN computational graphs, layer edges in message flows carry weights, and during evaluation, a single layer edge can appear in multiple message flows with different weights. However, GNN propagation rules require each layer edges to share a single weight, making it impossible to merge these flows while adhering to GNN propagation rules (see Figure 6 for an example). A mapping function is needed to convert the contributions of message flows to layer edges. Existing methods use average or sum functions as mapping functions, overlooking the fact that layer edges may contribute differently. To fairly attribute the contributions of message flows, we use the Shapley value.

For a message flow $\mathcal{F} = (\mathcal{F}[0],\ldots,\mathcal{F}[T])\in\Delta\mathcal{F}$, the corresponding layer edges in the $G_\tau$ can be represented as $\{a_{\mathcal{F}[\tau]\mathcal{F}[1]}^{\tau,1},\ldots,a_{\mathcal{F}[T-1]\mathcal{F}[T]}^{\tau,T}\}$, where $\tau\in\{0,1\}$. The changed layer edges in given message flow is $\Delta\mathcal{A}_\mathcal{F} = \{a_{\mathcal{F}[t-1]\mathcal{F}[t]}^t : a_{\mathcal{F}[t-1]\mathcal{F}[t]}^{0,t}\neq a_{\mathcal{F}[t-1]\mathcal{F}[t]}^{1,t}, t\in\{1,\ldots,T\}\}$. We consider mapping the contribution values $\mathbf{C}_s$, computed using Eq. (7), to the changed layer edges in $\Delta\mathcal{A}_\mathcal{F}$ as an allocation problem. We use the Shapley value $\phi_i = \sum_{S\subseteq N\setminus\{i\}}\frac{(|S|!(|N|-|S|-1)!)}{(|N|-1)!}(\nu(S\cup\{i\})-\nu(S))$ to fairly distribute $\mathbf{C}_s$ among these layer edges. We define the following:

- The player $i$: One changed layer edge $a_{\mathcal{F}[t-1]\mathcal{F}[t]}^t$ in $\Delta\mathcal{A}_\mathcal{F}$.

- The player sets $N$: $N = \Delta\mathcal{A}_\mathcal{F}$ denotes all changed layer edges in the message flow. $|N|$ is the total number of players.

- The coalition $S$: $S\subset N$. Only the weights of the layer edges in $S$ will be altered. This change will yield different layer edges for the given message flow. Consequently, the contribution of the same

message with different layer edges will differ. For given $S$, the corresponding layer edges are $\{\hat{a}^1_{\mathcal{F}[0]\mathcal{F}[1]}, \cdots \hat{a}^t_{\mathcal{F}[t-1]\mathcal{F}[t]} \cdots \hat{a}^T_{\mathcal{F}[T-1]\mathcal{F}[T]}\}$, where $\hat{a}^t_{\mathcal{F}[t-1]\mathcal{F}[t]} = a^{1,t}_{\mathcal{F}[t-1]\mathcal{F}[t]}$, if $a^t_{\mathcal{F}[t-1]\mathcal{F}[t]} \in S$, else $\hat{a}^t_{\mathcal{F}[t-1]\mathcal{F}[t]} = a^{0,t}_{\mathcal{F}[t-1]\mathcal{F}[t]}$. $|S|$ represents the size of $S$.

- $\nu(S)$: Given $S$ and the corresponding layer edges, $\nu(S)$ is computed according to Eq. (7) (node classification) or Eq. (13) (link prediction) or Eq. (14) (graph classification).

Given a message flow $\mathcal{F}$ in $\Delta\mathcal{F}$, we use the Shapley value to calculate the contribution $\phi_{a^t_{\mathcal{F}[t-1]\mathcal{F}[t]}}(\mathcal{F})$ of layer edge $a^t_{\mathcal{F}[t-1]\mathcal{F}[t]}$ to the message flow $\mathbf{C}_s$. Due to efficiency of Shapley value, it follows that $\sum_{a^t_{\mathcal{F}[t-1]\mathcal{F}[t]} \in N} \phi_{a^t_{\mathcal{F}[t-1]\mathcal{F}[t]}}(\mathcal{F}) = \nu(N) = \mathbf{C}_s$. An example of this calculation is illustrated in Figure 4.

For node classification, let the $\Phi \in \mathbb{R}^{|\Delta\mathcal{A}| \times c}$ denote the contribution matrix of layer edges, the row vector $\Phi_l$ denote the contribution of $l$-th layer edge $a^t_{\mathcal{F}[t-1]\mathcal{F}[t]}$ in $\Delta\mathcal{A}$. $\Phi_l = \sum_{\mathcal{F} \in \Delta\mathcal{F}} \phi_{a^t_{\mathcal{F}[t-1]\mathcal{F}[t]}}(\mathcal{F})$. Because $\sum_{a^t_{\mathcal{F}[t-1]\mathcal{F}[t]} \in N} \phi_{a^t_{\mathcal{F}[t-1]\mathcal{F}[t]}}(\mathcal{F}) = \mathbf{C}_s$ and $\sum_{s=1}^{|\Delta\mathcal{F}|} \mathbf{C}_s = \Delta\mathbf{z}_J$, $\sum_{l=1}^{|\Delta\mathcal{A}|} \Phi_l = \Delta\mathbf{z}_J$. For link prediction and graph classification tasks, see Appendix A.3.

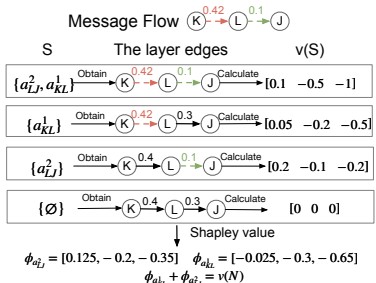

Figure 4: For message flow in $G_1$ in Figure 1, the example of using Shapley value to map contribution from message flows to layer edges.

## 3.3 SELECT THE IMPORTANT LAYER EDGES

To illustrate the selection of important layer edges, we focus on node classification task, while link prediction and graph classification tasks are detailed in the Appendix A.4. KL-divergence is used to measure the approximation quality of a static predicted distribution $\mathsf{Pr}_J(G)$ (Liu et al., 2024):

$$\mathsf{KL}(\mathsf{Pr}_J(G_1) \| \mathsf{Pr}_J(G_0)) = \sum_{k=1}^{c} \mathsf{Pr}_k(G_1) \log[\mathsf{Pr}_k(G_1)/\mathsf{Pr}_k(G_0)]$$

$$= \sum_{k=1}^{c} \mathsf{Pr}_k(G_1)[z_k(G_1) - z_k(G_0)] - \log[\frac{Z(G_1)}{Z(G_0)}] = \sum_{k=1}^{c} \mathsf{Pr}_k(G_1)\Delta z_k - \log[\frac{Z(G_1)}{Z(G_0)}], \quad (8)$$

where $Z(G_\tau) = \sum_{k=1}^{c} \exp(z_k(G_\tau))$ for $\tau = 0, 1$. Let $\mathbf{x} \in \{0, 1\}^{|\Delta\mathcal{A}|}$ denote the selection vector, where the element $x_l$ in $\mathbf{x}$ indicates whether the $l$-th layer edge is selected. Suppose we select a subset of $n$ important changed layer edges $\Delta\mathcal{A}_{sub} \in \Delta\mathcal{A}$ to explain evolution from $\mathsf{Pr}(Y|G_0; \boldsymbol{\theta})$ to $\mathsf{Pr}(Y|G_1; \boldsymbol{\theta})$. Let $G_n$ denote the graph that the weights of the layer edges in $\Delta\mathcal{A}_{sub}$ are changed to those in $G_1$, while the weights of the layer edges in $\Delta\mathcal{A} \setminus \Delta\mathcal{A}_{sub}$ remain unchanged. If the $\mathsf{KL}(\mathsf{Pr}(G_1) \| \mathsf{Pr}(G_n))$ is small, it means that the $\Delta\mathcal{A}_{sub}$ can faithfully explain the evolution in prediction probability. Let $\Phi$ denote the contribution matrix of layer edges, and $\Phi_l$ represent the contribution of $l$-th layer edge to $\Delta\mathbf{z}_J$. $\Phi_{l,k}$ indicates the contribution of $l$-th layer edge to $\Delta z_k$. Thus, $\mathbf{z}_J(G_n) = \sum_{l=1}^{|\Delta\mathcal{A}|} x_l\Phi_l + \mathbf{z}_J(G_0)$, and according to Eq. (8), $\mathsf{KL}(\mathsf{Pr}(G_1) \| \mathsf{Pr}(G_n)) = \sum_{k=1}^{c} \mathsf{Pr}_k(G_1)\big(z_k(G_1) - z_k(G_n)\big) - \log Z(G_1) + \log Z(G_n)$, where $Z(G_n) = \sum_{k=1}^{c} \exp(z_k(G_n))$, then we can define the following objective function:

$$\mathbf{x}^* = \underset{\substack{\mathbf{x} \in \{0,1\}^{|\Delta\mathcal{A}|} \\ \|\mathbf{x}\|_1 = n}}{\arg\min} \sum_{k=1}^{c} \left( -\mathsf{Pr}_k(G_1) \sum_{l=1}^{|\Delta\mathcal{A}|} x_l\Phi_{l,k} \right) + \log \sum_{k'=1}^{c} \exp \left( z_{k'}(G_0) + \sum_{l=1}^{|\Delta\mathcal{A}|} x_l\Phi_{l,k'} \right)$$

$$(9)$$

By solving Eq. (9), we can obtain the most important changed layer edges. The Algorithm 1 shows the overall process of selecting important layer edges for node classification task. The Algorithm 2 for the link prediction task and Algorithm 3 for the graph classification task are in the Appendix.

---

**Algorithm 1** Selecting important layer edges to explain evolution of $\Pr(Y|G_0)$ to $\Pr(Y|G_1)$ on the node classification task

---

1: **Input**: the source graph $G_0$ and the destination graph $G_1$, Pre-trained GNN parameters $\boldsymbol{\theta}$
2: Obtain the layer edges flow set $\Delta\mathcal{A}$
3: Initialize layer edges contribution matrix $\Phi \in \mathbb{R}^{|\Delta\mathcal{A}| \times c}$ as an all-zero matrix
4: Obtain the altered massage flows set $\Delta\mathcal{F}$
5: Given the target node $J$, $\Delta\mathcal{F} = \{\mathcal{F} : \mathcal{F} \in \Delta\mathcal{F} \text{ and } \mathcal{F}[T] = J\}$
6: **for** $s$ for 1 to $|\Delta\mathcal{F}|$ **do**
7:     Select the $s$-th message flow in $|\Delta\mathcal{F}|$ and calculate contribution $\mathbf{C}_s$ according to the Eq. (7)
8:     Obtain the changed layer edges set $\Delta\mathcal{A}_\mathcal{F}$ on this flow
9:     **for** $a^t_{\mathcal{F}[t-1]\mathcal{F}[t]}$ in $\Delta\mathcal{A}_\mathcal{F}$ **do**
10:         Calculate $\phi_{a^t_{\mathcal{F}[t-1]\mathcal{F}[t]}}(\mathcal{F})$ using Shapley value.
11:         Let the index of $a^t_{\mathcal{F}[t-1]\mathcal{F}[t]}$ in $\Delta\mathcal{A}$ is $l$, $\Phi_l = \Phi_l + \phi_{a^t_{\mathcal{F}[t-1]\mathcal{F}[t]}}(\mathcal{F})$
12:     **end for**
13: **end for**
14: Solve Eq. (9) to obtain the important changed layer edges
15: **Output**: The important changed layer edges set

---

### 3.4 Complexity Analysis

**Obtain the changed message flows $\Delta\mathcal{F}$:** Given the changed edges $\Delta\mathcal{E}$, we use the depth-first search method to obtain $\Delta\mathcal{F}$ and the complexity is $O\big(|\Delta\mathcal{E}|^T\big)$.

**Calculate contributions:** According to the Eq. (7), we calculate the contribution of each message flow through the vectorized method. The complexity is $O\big(|\Delta\mathcal{F}|d^1 \cdots d^t d^{t+1} \cdots d^{T+1}\big)$, where $d^t$ and $d^{t+1}$ denote the dimension of the $\boldsymbol{\theta}^t \in \mathbb{R}^{d^t \times d^{t+1}}, t = \{1, \cdots, T\}$.

**Apply the Shapley value:** For each message flow $\mathcal{F} \in \Delta\mathcal{F}$ with more than one changed layer edge, the Shapley value is used to fairly attribute contributions. Some calculations in the calculating contributions can be used repeatedly. The worst-case complexity is $O\big(|\Delta\mathcal{F}|(2^T - 1)d^1 \cdots d^{T+1}\big)$, where $(2^T - 1)$ represents the number of non-empty subsets of layers.

**Select the inportant layer edges:** The time complexity is $O\big(|\Delta\mathcal{A}|^3\big)$.

## 4 Experiment

**Datasets and tasks**. We evaluate our method on node classification, link prediction and graph classification tasks using real and simulated dynamic graph datasets. Details of these datasets are provided in Appendix A.7.1. We assess the running time of our method on large datasets (See Appendix A.7.7 and Figure 9). On BA-Shapes dataset, we validate the accuracy of the explanation methods. The visualization results and accuracy are shown in Appendix A.7.8, Figures 10 and 11.

**Experimental setup**. For each dataset, we optimize the GNN parameter $\boldsymbol{\theta}$ on the training set of static graphs, using labeled nodes, edges, or graphs based on the specific tasks. For each graph snapshot, excluding the first one, target nodes/edges/graphs with a significantly large $D_{\text{KL}}(\Pr(Y|G_0)||\Pr(Y|G_1))$ are collected and the change in $\Pr(Y|G)$ is explained. We run Algorithm 1 or Algorithm 3 to identify the important layer edges for node classification, link prediction and graph classification tasks. The optimization problems in Eq. (9), Eq. (15) and Eq. (16) are solved using the cvxpy library (Diamond & Boyd). Our proposed method is called "**AxiomLayeredge**". Additionally, we employ the GNNExplainer (Ying et al., 2019), PG-Explainer (Luo et al., 2020), GNNLRP (Schnake et al., 2020) , DeeoLIFT (Shrikumar et al., 2017), and the FlowX (Gui et al., 2023) as our baselines. We also design some variant methods: AxiomLayeredge-Topk, AxiomLayeredge\Shapley, AxiomEdge, AxiomEdge-Top and AxiomEdge\Shapley, Appendix A.7.2 gives details of the baseline methods. Appendix A.7.3 gives details of the experimental setup.

**Quantitative evaluation metrics**. Supposing the selected important layer edges set and edges set are denoted as $\Delta\mathcal{A}^*$ and $\Delta\mathcal{E}^*$, respectively. For evaluating the layer edges, we start from the computational graph of $G_0$, only adjusting the weights of the layer edges in $\Delta\mathcal{A}^*$ to those $G_1$, while the weights of the layer edges in $\mathcal{A} \setminus \Delta\mathcal{A}^*$ remain unchanged, then we obtain the computational graph $G_n$. Similarly, for edges evaluation, we alter the weights of the edges in $\Delta\mathcal{E}^*$ to those in $G_1$, with other edges weights in $\mathcal{E} \setminus \Delta\mathcal{E}^*$ unchanged, then we also obtain the $G_n$. After obtaining $G_n$, we can compute the $\Pr(Y|G(n))$. The case of obtaining $G_n$ can be seen in Figure 7.

The evaluation metric for the node classification is Kullback-Leibler (KL) divergence $\mathrm{KL}(\Pr_J(G_1)\|\Pr_J(G_n))$. See Figure 7 for an example. The idea of this metric is similar to the Fidelity- (Yuan et al., 2020b). Intuitively, if adjusting only the weights of selected layer edges (rather than all changed edges) brings $\Pr_J(G_n)$ closer to $\Pr_J(G_1)$, it indicates that these edges effectively explain the evolution from $\Pr(Y|G_0)$ to $\Pr(Y|G_1)$, resulting in a smaller evaluation metric. A similar metric can be defined for the link prediction task and the graph classification task, where the KL-divergence is calculated using predicted distributions over the target edge or graph. To ensure comparability between layer edges based and edges based explanations, we apply the same level of sparsity. We define five levels of explanation sparsity (Yuan et al., 2020b), with all methods compared under the same sparsity level. For the edges, the sparsity is $1 - \frac{\Delta\mathcal{E}^*}{\Delta\mathcal{E}}$. For the layer edges, it is $1 - \frac{\Delta\mathcal{A}^*}{\Delta\mathcal{A}}$. The higher sparsity indicates the explanations are more sparse and tend to only capture the most important input information. The Table 4 and Table 5 in Appendix A.7.4 provide details on the sparsity for real and simulated dynamic graph datasets across the three graph tasks.

**Performance evaluation and comparison**. We evaluate the performance of the methods across three tasks: node classification, link prediction and graph classification in real and simulated dynamic graph datasets. For each dataset, we report the average KL over target nodes/edges/graphs. Results for dynamic graph datasets are illustrated in Figure 5, while those for simulated dynamic graphs are presented in Figure 8 in Appendix A.7.6. Table 4 and Table 5 display explanation sparsity levels across different datasets. In Table 4, the sparsity for all real dynamic graph datasets is no less than 0.9. Figure 5 demonstrates that our method AxiomLayeredge has the smallest KL across all levels of explanation sparsity levels, datasets, and tasks, with exception of certain sparsity levels of Pheme dataset. This illustrates that our method maintains high fidelity in explanations even under high sparsity. On eight settings (Weibo, YelpChi, YelpNYC, BC-Alpha, BC-OTC, UCI, MUTAG, ClinTox), our method AxiomLayeredge along with its variants AxiomEdge, AxiomEdge\Shapley, AxiomLayeredge\Shapley outperform the GNNLRP, DeepLIFT, GNNExplainer, PGExplainer and FlowX methods. This demonstrates that our proposed methods more effectively explain the evolution of $\Pr(Y|G_0;\boldsymbol{\theta})$ to $\Pr(Y|G_1;\boldsymbol{\theta})$, while methods designed for static graph struggle to identify salient edges that explain changes in the predicted probability distribution. Moreover, our method AxiomLayeredge has superior performance compared to the AxiomLayeredge\Shapley method across all levels of explanation sparsity, datasets, and tasks, with a significant gap observed on the Pheme and Weibo datasets. Therefore, the Shapley value provides a fair attribution.

## 5 RELATED WORK AND FURTHER DISCUSSION

**GNNs Explainability.** The limitation of GNNs is the lack of explainability. Recently, various methods have been proposed to explain GNN predictions, primarily focusing on static graphs. In the survey (Yuan et al., 2020b), existing GNN explanation approaches are categorized as instance-level and model-level methods. The instance-level category includes **Gradient/features-based** methods, such as CAM and GradCAM (Baldassarre & Azizpour, 2019; Pope et al., 2019), which identify important nodes by the gradient,but are not applicable for the node classification. **Perturbation-based** methods, such as GNNexplainer (Ying et al., 2019), PGExplainer (Luo et al., 2020), GraphMask (Schlichtkrull et al., 2020), learn masks to identify important edges by maximizing the mutual information to explain the predicted class distribution of model. However, these methods cannot axiomatically isolate contributions of message flows that causally impact the prediction changes on the computation graphs. **Decomposition-based** methods, such as GNN-LRP (Schnake et al., 2020), extend the original LRP (Bach et al., 2015) algorithm to GNNs and study the importance of the graph walks. While GNN-LRP explains the single class probability, it cannot explain multi-class distributions change over evolving graphs. **Surrogate-based** methods, like GraphLime (Huang et al., 2020), use a surrogate model with kernel-based feature selection to provide node feature explanations. In model-level category, XGNN (Yuan et al., 2020a) generates graph patterns that maximize

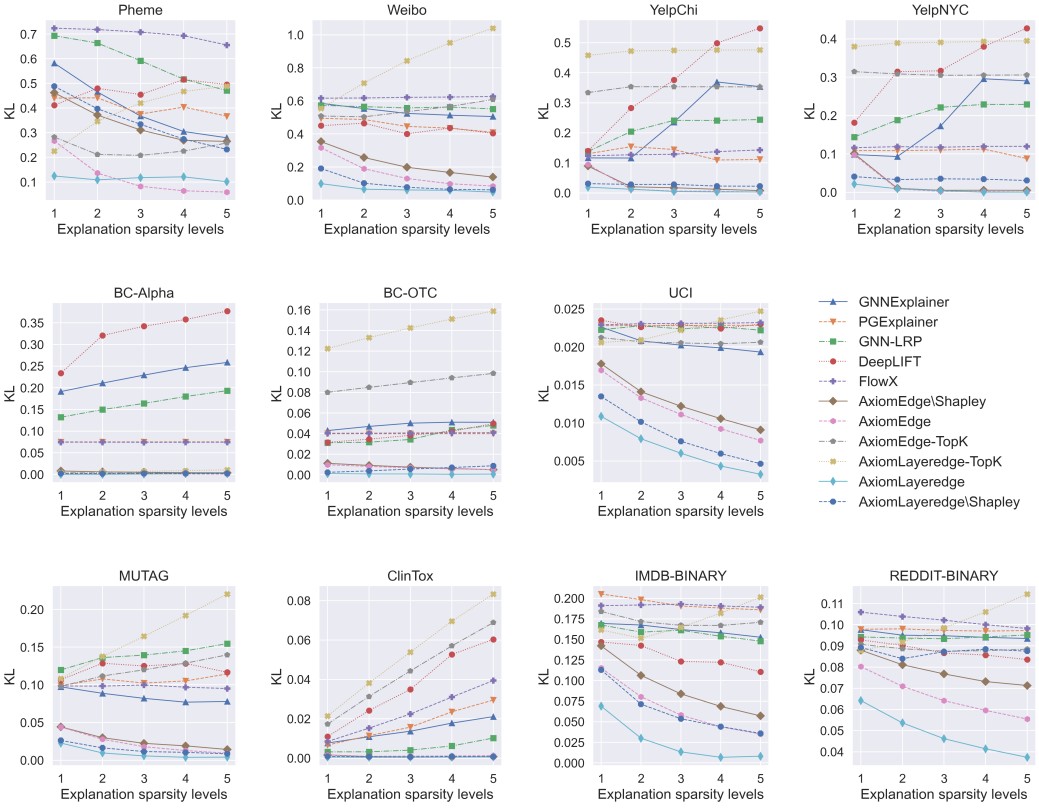

Figure 5: Performance of KL in real dynamic graphs. Each figure corresponds to a different dataset. The first, second and third rows represent node classification, link prediction and graph classification tasks, respectively.

a certain model prediction via reinforcement learning. In conclusion, most prior work evaluate the fidelity of the explanations on static graphs. They cannot explain the evolution of $\Pr(Y|G_0; \boldsymbol{\theta})$ to $\Pr(Y|G_1; \boldsymbol{\theta})$.

**Limitations and Concerns.** While we acknowledge the effectiveness of our methods, we also recognize their limitations. We focus on the explanations on static GNNs in evolving graphs. Our method cannot be extended to dynamic graph models, such as TGN (Rossi et al., 2020), which incorporate time-series components like RNNs or LSTMs. We do not design rules for decomposing contributions in RNNs or LSTMs. For static GNNs, our method can be applied to the GIN (Xu et al., 2018) model. For GAT model, our explanations are limited to identifying a small subset of changed attention weights on edges. We are unable to measure the contribution of added or removed edges to the changed attention weights and select the changed edges as the explanations. Because we do not define the rules to attribute the difference of softmax function to the changed input.

# 6 CONCLUSIONS

We studied the problem of explaining change in GNN predictions with the weights of input edges continuously changed. We addressed the issues of prior works, such as lack of axiomatic attribution of message flows, unfair distribution and lack of optimality. The proposed algorithm can axiomatically decompose the changes to message flow in the computation graphs of GNN and employ the Shapley value for fair attribution to layer edges. It further optimally select a small subset of layer edges to explain the evolution of prediction probabilities. Experimental results demonstrate that our method achieves superior performance even when sparsity exceeds 0.9. This indicates that our approach successfully balances Interpretability and Fidelity.

## 7 ACKNOWLEDGEMENT

Sihong Xie was supported by the Department of Science and Technology of Guangdong Province (2023CX10X079), National Key R&D Program of China (Grant No.2023YFF0725001), the Guangzhou-HKUST(GZ) Joint Funding Program (Grant No.2023A03J0008), and Education Bureau Guangzhou Municipality.

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

## A    APPENDIX

### A.1    EVALUATION OF MESSAGE FLOWS ON DYNAMIC GRAPHS

In Figure 6, we illustrate the computation of Fidelity for both dynamic and static graphs from the perspective of computational graphs. The static graph $G_0$ is considered an evolution of $G_{\text{empty}}$. In the case of dynamic graphs, $G_1$ evolves from the $G_0$. After identifying the important message flows, we adjust their weights to align with those in the destination graph, keeping the weights of the remaining flows unchanged. This process generates a new computational graph $G_n$. In dynamic graphs, adjusting the weights of selected important message flows may lead to differing weights for the same-layer edges across various flows. However, GNN propagation rules require that edges within each layer share a single weight. Thus, merging these flows while complying with GNN propagation constraints is infeasible.

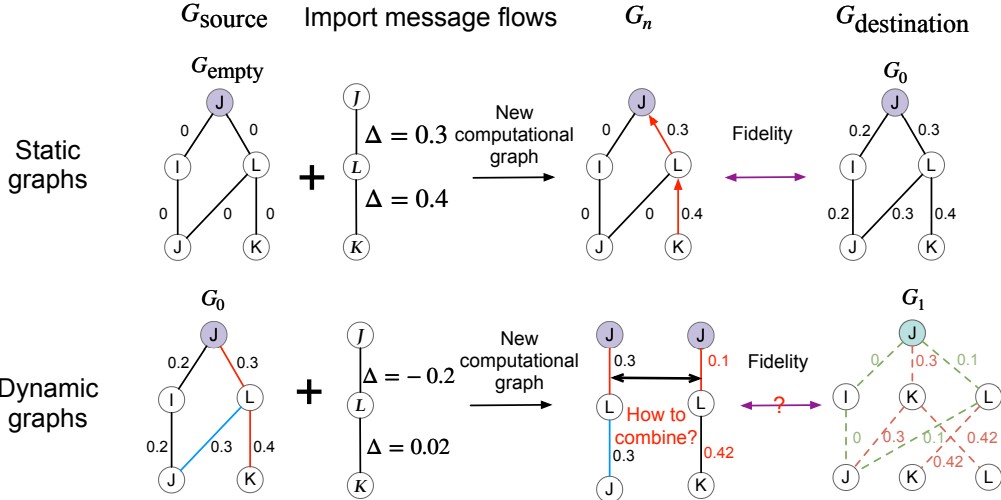

Figure 6: Calculation of Fidelity for dynamic and static graphs. Challenges may arise during the computation for dynamic graphs.

### A.2    CALCULATE THE CONTRIBUTION OF MESSAGE FLOWS

#### A.2.1    THE EXAMPLES ON THE NODE PREDICTION TASKS

Supposing the GNN models have two layers, considering the massage flow $\mathcal{F} = (V, I, J) \in$ the altered message flows set $\Delta\mathcal{F}$, We have derived in detail the calculation process of the contribution value of message flow:

$$
\begin{aligned}
\mathbf{C}_s &= a_{IJ}^{0,T}\Delta\mathbf{h}_I^{t-1}\boldsymbol{\theta}^T + \Delta a_{IJ}^t\mathbf{h}_I^{1,t-1}\boldsymbol{\theta}^T \quad \text{the contribution of } \Delta\mathbf{h}_I,\ \mathbf{h}_I \text{ to } \Delta\mathbf{z}_J \\
&= a_{IJ}^{0,T}\left(\Delta\mathbf{z}_I^{T-1}\mathbf{m}_{\Delta\mathbf{z}_I^{T-1}\Delta\mathbf{h}_I^{T-1}}\right)\boldsymbol{\theta}^T \quad \text{the contribution of } \Delta\mathbf{z}_I \text{ to } \Delta\mathbf{h}_I \\
&\quad + \Delta a_{IJ}^T\left(\mathbf{z}_I^{1,T-1}\mathbf{m}_{\mathbf{z}_I^{1,T-1}\mathbf{h}_I^{1,T-1}}\right)\boldsymbol{\theta}^T \quad \text{the contribution of } \mathbf{z}_I \text{ to } \mathbf{h}_I \\
&= a_{IJ}^{0,T}\Delta\mathbf{h}_V^{T-2}\mathbf{m}_{\Delta\mathbf{h}_V^{T-2}\Delta\mathbf{z}_I^{T-1}}\mathbf{m}_{\Delta\mathbf{z}_I^{T-1}\Delta\mathbf{h}_I^{T-1}}\boldsymbol{\theta}^T \quad \text{the contribution of } \Delta\mathbf{h}_V \text{ to } \Delta\mathbf{z}_J \\
&\quad + a_{IJ}^{0,T}\mathbf{h}_V^{1,T-2}\mathbf{m}_{\mathbf{h}_V^{1,T-2}\Delta\mathbf{z}_I^{T-1}}\mathbf{m}_{\Delta\mathbf{z}_I^{T-1}\Delta\mathbf{h}_I^{T-1}}\boldsymbol{\theta}^T \quad \text{the contribution of } \mathbf{h}_V \text{ to } \Delta\mathbf{z}_J \\
&\quad + \Delta a_{IJ}^T\left(\mathbf{h}_V^{1,T-2}\mathbf{m}_{\mathbf{h}_V^{1,T-2}\mathbf{z}_I^{1,T-1}}\right)\mathbf{m}_{\mathbf{z}_I^{1,T-1}\mathbf{h}_I^{1,T-1}}\boldsymbol{\theta}^T \quad \text{the contribution of } \mathbf{h}_V \text{ to } \Delta\mathbf{z}_J
\end{aligned}
\tag{10}
$$

According to the multiplier designed by the DeepLIFT, $\mathbf{m}_{\Delta\mathbf{h}_V^{T-2}\Delta\mathbf{z}_I^{T-1}} =$ $\Delta a_{VI}^{T-1}\boldsymbol{\theta}^{T-1}, \mathbf{m}_{\Delta\mathbf{z}_I^{T-1}\Delta\mathbf{h}_I^{T-1}} = \frac{\Delta\mathbf{h}_I^{T-1}}{\Delta\mathbf{z}_I^{T-1}}, \mathbf{m}_{\mathbf{z}_I^{T-1}\mathbf{h}_I^{T-1}} = \frac{\mathbf{h}_I^{T-1}}{\mathbf{z}_I^{T-1}}, \mathbf{m}_{\mathbf{h}_V^{1,T-2}\mathbf{z}_I^{1,T-1}} = a_{VI}^{1,T-1}\boldsymbol{\theta}^{T-1},$

therefore,

$$\mathbf{C}_s = \Delta a_{VI}^{T-1} a_{IJ}^{0,T} \mathbf{h}_V^{1,T-2} \boldsymbol{\theta}^{T-1} \frac{\Delta \mathbf{h}_I^{T-1}}{\Delta \mathbf{z}_I^{T-1}} \boldsymbol{\theta}^T + a_{VI}^{1,T-1} \Delta a_{IJ}^T \mathbf{h}_V^{1,T-2} \boldsymbol{\theta}^{T-1} \frac{\mathbf{h}_I^{T-1}}{\mathbf{z}_I^{T-1}} \boldsymbol{\theta}^T \quad (11)$$

Where the divide means the element-wise division, $T = 2$.

Similarly, Supposing the GNN models have three layers, considering the massage flow $\mathcal{F} = (U, V, I, J) \in$ the altered message flows set $\Delta \mathcal{F}$, We have derived in detail the calculation process of the contribution value of message flow:

$$
\begin{aligned}
\mathbf{C}_s &= a_{IJ}^{0,T} \Delta \mathbf{h}_I^{t-1} \boldsymbol{\theta}^T + \Delta a_{IJ}^t \mathbf{h}_I^{1,t-1} \boldsymbol{\theta}^T \quad \text{the contribution of } \Delta \mathbf{h}_I, \mathbf{h}_I \text{ to } \Delta \mathbf{z}_J \\
&= a_{IJ}^{0,T} \big(\Delta \mathbf{z}_I^{T-1} \mathbf{m}_{\Delta \mathbf{z}_I^{T-1} \Delta \mathbf{h}_I^{T-1}}\big) \boldsymbol{\theta}^T \quad \text{the contribution of } \Delta \mathbf{z}_I \text{ to } \Delta \mathbf{h}_I \\
&\quad + \Delta a_{IJ}^T \big(\mathbf{z}_I^{1,T-1} \mathbf{m}_{\mathbf{z}_I^{1,T-1} \mathbf{h}_I^{1,T-1}}\big) \boldsymbol{\theta}^T \quad \text{the contribution of } \mathbf{z}_I \text{ to } \mathbf{h}_I \\
&= a_{IJ}^{0,T} \Delta \mathbf{h}_V^{T-2} \mathbf{m}_{\Delta \mathbf{h}_V^{T-2} \Delta \mathbf{z}_I^{T-1}} \mathbf{m}_{\Delta \mathbf{z}_I^{T-1} \Delta \mathbf{h}_I^{T-1}} \boldsymbol{\theta}^T \quad \text{the contribution of } \Delta \mathbf{h}_V \text{ to } \Delta \mathbf{z}_J \\
&\quad + a_{IJ}^{0,T} \mathbf{h}_V^{1,T-2} \mathbf{m}_{\mathbf{h}_V^{1,T-2} \Delta \mathbf{z}_I^{T-1}} \mathbf{m}_{\Delta \mathbf{z}_I^{T-1} \Delta \mathbf{h}_I^{T-1}} \boldsymbol{\theta}^T \quad \text{the contribution of } \mathbf{h}_V \text{ to } \Delta \mathbf{z}_J \\
&\quad + \Delta a_{IJ}^T \big(\mathbf{h}_V^{1,T-2} \mathbf{m}_{\mathbf{h}_V^{1,T-2} \mathbf{z}_I^{1,T-1}}\big) \mathbf{m}_{\mathbf{z}_I^{1,T-1} \mathbf{h}_I^{1,T-1}} \boldsymbol{\theta}^T \quad \text{the contribution of } \mathbf{h}_V \text{ to } \Delta \mathbf{z}_J \\
&= a_{IJ}^{0,T} \big(a_{UV}^{0,T-2} \Delta \mathbf{h}_U^{T-3} \boldsymbol{\theta}^{T-2} + \Delta a_{UV}^{T-2} \mathbf{h}_U^{1,T-3} \boldsymbol{\theta}^{T-2}\big) \mathbf{m}_{\Delta \mathbf{h}_V^{T-2} \Delta \mathbf{z}_I^{T-1}} \mathbf{m}_{\Delta \mathbf{z}_I^{T-1} \Delta \mathbf{h}_I^{T-1}} \boldsymbol{\theta}^T \\
&\quad \text{the contribution of } \Delta \mathbf{h}_U \text{ to } \Delta \mathbf{z}_J \\
&\quad + a_{IJ}^{0,T} \big(\mathbf{h}_U^{T-3} \mathbf{m}_{\mathbf{h}_U^{1,T-3} \mathbf{z}_V^{1,T-2}} \mathbf{m}_{\mathbf{z}_V^{1,T-2} \mathbf{h}_V^{1,T-2}}\big) \mathbf{m}_{\mathbf{h}_V^{1,T-2} \Delta \mathbf{z}_I^{T-1}} \mathbf{m}_{\Delta \mathbf{z}_I^{T-1} \Delta \mathbf{h}_I^{T-1}} \boldsymbol{\theta}^T \quad (12) \\
&\quad \text{the contribution of } \mathbf{h}_U \text{ to } \Delta \mathbf{z}_J \\
&\quad + \Delta a_{IJ}^T \big(\mathbf{h}_U^{T-3} \mathbf{m}_{\mathbf{h}_U^{1,T-3} \mathbf{z}_V^{1,T-2}} \mathbf{m}_{\mathbf{z}_V^{1,T-2} \mathbf{h}_V^{1,T-2}}\big) \mathbf{m}_{\mathbf{h}_V^{1,T-2} \mathbf{z}_I^{1,T-1}} \mathbf{m}_{\mathbf{z}_I^{1,T-1} \mathbf{h}_I^{1,T-1}} \boldsymbol{\theta}^T \\
&\quad \text{the contribution of } \mathbf{h}_U \text{ to } \Delta \mathbf{z}_J \\
&= \Delta a_{UV}^{0,T-2} a_{VI}^{0,T-1} a_{IJ}^{0,T} \mathbf{h}_U^{1,T-3} \boldsymbol{\theta}^{T-2} \frac{\Delta \mathbf{h}_V^{T-2}}{\Delta \mathbf{z}_V^{T-2}} \boldsymbol{\theta}^{T-1} \frac{\Delta \mathbf{h}_I^{T-1}}{\Delta \mathbf{z}_I^{T-1}} \boldsymbol{\theta}^T \\
&\quad + a_{UV}^{1,T-2} \Delta a_{VI}^{T-1} a_{IJ}^{0,T} \mathbf{h}_U^{1,T-3} \boldsymbol{\theta}^{T-2} \frac{\mathbf{h}_V^{T-2}}{\mathbf{z}_V^{T-2}} \boldsymbol{\theta}^{T-1} \frac{\Delta \mathbf{h}_I^{T-1}}{\Delta \mathbf{z}_I^{T-1}} \boldsymbol{\theta}^T \\
&\quad + a_{UV}^{1,T-2} a_{VI}^{1,T-1} \Delta a_{IJ}^T \mathbf{h}_U^{1,T-3} \boldsymbol{\theta}^{T-2} \frac{\mathbf{h}_V^{T-2}}{\mathbf{z}_V^{T-2}} \boldsymbol{\theta}^{T-1} \frac{\mathbf{h}_I^{T-1}}{\mathbf{z}_I^{T-1}} \boldsymbol{\theta}^T
\end{aligned}
$$

### A.2.2 ON THE LINK PREDICTION TASK

According to Eq. (3), for the target edge $e_{IJ}$, the $\mathbf{z}_I^T \in \mathbb{R}^{1 \times d}$ and $\mathbf{z}_J^T \in \mathbb{R}^{1 \times d}$ are concatenated and passed through a linear layer with the parameters $\boldsymbol{\theta}_{LP}$. According to Eq. (7), we can obtain the contribution of message flow to $\Delta \mathbf{z}_I^T$ or $\Delta \mathbf{z}_J^T$, then the contribution of message flow to the $\Delta \mathbf{z}_{IJ} = \mathbf{z}_{IJ}(G_1) - \mathbf{z}_{IJ}(G_0)$ is:

$$
\begin{aligned}
\mathbf{C_s} = \sum_{t=0}^{T-1} \Big( & a_{\mathcal{F}[0]\mathcal{F}[1]}^{1,1} a_{\mathcal{F}[1]\mathcal{F}[2]}^{1,2} \cdots \Delta a_{\mathcal{F}[t]\mathcal{F}[t+1]}^{t+1} a_{\mathcal{F}[t+1],\mathcal{F}[t+2]}^{0,t+2} \cdots a_{\mathcal{F}[T-1],\mathcal{F}[T]}^{0,T} \\
& \mathbf{h}_{\mathcal{F}[0]}^{1,0} \frac{\mathbf{h}_{\mathcal{F}[1]}^{1,1}}{\mathbf{z}_{\mathcal{F}[1]}^{1,1}} \cdots \frac{\mathbf{h}_{\mathcal{F}[t]}^{1,t}}{\mathbf{z}_{\mathcal{F}[t]}^{1,t}} \boldsymbol{\theta}^t \frac{\Delta \mathbf{h}_{\mathcal{F}[t+1]}^{t+1}}{\Delta \mathbf{z}_{\mathcal{F}[t+1]}^{t+1}} \boldsymbol{\theta}^{t+1} \cdots \frac{\Delta \mathbf{h}_{\mathcal{F}[T-1]}^{T-1}}{\Delta \mathbf{z}_{\mathcal{F}[T-1]}^{T-1}} \boldsymbol{\theta}^T \boldsymbol{\theta}_{LP}' \Big)
\end{aligned} \quad (13)
$$

Where $\boldsymbol{\theta}_{LP}' = \boldsymbol{\theta}_{LP}[0 : d]$, $d$ if $V_{T+1} = I$, $\boldsymbol{\theta}_{LP}' = \boldsymbol{\theta}_{LP}[d :]$, if $V_{T+1} = J$

### A.2.3 ON THE GRAPH CLASSIFICATION TASK

Because the average pooling is used for the graph classification tasks, $\Delta \mathbf{z} = \mathbf{z}(G_1) - \mathbf{z}(G_0) = \sum_{J \in (\mathcal{V}^0 \cup \mathcal{V}^1)} \Delta \mathbf{z}_J^T / |\mathcal{V}^0 \cup \mathcal{V}^1|$, thus the contribution is:

$$
\mathbf{C}_s = \sum_{t=0}^{T-1} \left( a^{1,1}_{\mathcal{F}[0]\mathcal{F}[1]} a^{1,2}_{\mathcal{F}[1]\mathcal{F}[2]} \cdots \Delta a^{t+1}_{\mathcal{F}[t]\mathcal{F}[t+1]} a^{0,t+2}_{\mathcal{F}[t+1],\mathcal{F}[t+2]} \cdots a^{0,T}_{\mathcal{F}[T-1],\mathcal{F}[T]} \right.
$$
$$
\left. \mathbf{h}^{1,0}_{\mathcal{F}[0]} \frac{\mathbf{h}^{1,1}_{\mathcal{F}[1]}}{\mathbf{z}^{1,1}_{\mathcal{F}[1]}} \cdots \frac{\mathbf{h}^{1,t}_{\mathcal{F}[t]}}{\mathbf{z}^{1,t}_{\mathcal{F}[t]}} \boldsymbol{\theta}^t \frac{\Delta \mathbf{h}^{t+1}_{\mathcal{F}[t+1]}}{\Delta \mathbf{z}^{t+1}_{\mathcal{F}[t+1]}} \boldsymbol{\theta}^{t+1} \cdots \frac{\Delta \mathbf{h}^{T-1}_{\mathcal{F}[T-1]}}{\Delta \mathbf{z}^{T-1}_{\mathcal{F}[T-1]}} \boldsymbol{\theta}^T \right) / |\mathcal{V}^0 \cup \mathcal{V}^1| \tag{14}
$$

Where, $\mathcal{V}_0$ and $\mathcal{V}_1$ denote the the nodes set of graph $G_0$ and $G_1$, respectively.

### A.3 MAPPING CONTRIBUTIONS FOR THE GRAPH CLASSIFICATION TASK

In the section 3.2, we show how to calculate the Shapley value, i.e. contribution $\phi_{a^t_{\mathcal{F}[t-1]\mathcal{F}[t]}}(\mathcal{F})$ of layer edge $a^t_{\mathcal{F}[t-1]\mathcal{F}[t]}$ to $\Delta \mathbf{z}^T_{\mathcal{F}_T}$. Note that the changed layer edge can affect many nodes, not the single node. Thus, in the graph classification task, the contribution matrix of $l$-th layer edge $a^t_{\mathcal{F}[t-1]\mathcal{F}[t]} \in \Delta \mathcal{A}$ is $\Phi^l \in \mathbb{R}^{|\mathcal{V}^0 \cup \mathcal{V}^1| \times c}$, the row vector $\Phi^l_i = \phi_{a^t_{\mathcal{F}[t-1]\mathcal{F}[t]}}(\mathcal{F})$ denotes the contribution of the $l$-th layer edge to $\Delta \mathbf{z}^T_{\mathcal{F}_T}$, where the $i$-th node in the $\mathcal{V}^0 \cup \mathcal{V}^1$ is $\mathcal{F}_T$. Let $\Phi = \sum_{l=1}^{|\Delta \mathcal{A}|} \Phi^l$, the $\Phi$ also follows the summation-to-delta property $\sum_{i=1}^{|\mathcal{V}^0 \cup \mathcal{V}^1|} \Phi_i = \Delta \mathbf{z} = \mathbf{z}(G_1) - \mathbf{z}(G_0)$

### A.4 SELECTING THE IMPORTANT LAYER EDGES

#### A.4.1 ON THE LINK PREDICTION TASK

For the link prediction, the $\mathbf{z}_{IJ}(G) = [\mathbf{z}_1, \cdots, \mathbf{z}_\ell \cdots, \mathbf{z}_c], \mathsf{Pr}_{IJ}(G) = [\mathsf{Pr}_1(G), \cdots, \mathsf{Pr}_\ell \cdots, \mathsf{Pr}_c(G)]$, Let $\Phi$ denotes the contribution matrix of layer edges, where $\Phi_l$ represents the contribution of $l$-th layer edge to $\Delta \mathbf{z}_{IJ}$, and $\Phi_{l,\ell}$ indicates the contribution of $l$-th layer edge to $\Delta z_\ell$, we can define the following objective function for the link prediction:

$$
\mathbf{x}^* = \underset{\substack{\mathbf{x} \in \{0,1\}^{|\Delta \mathcal{A}|} \\ \|\mathbf{x}\|_1 = n}}{\arg \min} \sum_{\ell=1}^{c} \left( -\mathsf{Pr}_\ell(G_1) \sum_{l=1}^{|\Delta \mathcal{A}|} x_l \Phi_{l,\ell} \right)
$$
$$
+ \log \sum_{\ell'=1}^{c} \exp \left( z_{\ell'}(G_0) + \sum_{l=1}^{|\Delta \mathcal{A}|} x_l \Phi_{l,\ell'} \right) \tag{15}
$$

#### A.4.2 ON THE GRAPH CLASSIFICATION TASK

For the graph classification, the $\Phi^l$ denotes contribution matrix of the $l$-th layer edge in the $\Delta \mathcal{A}$. The logits of the graph classification $\mathbf{z}_G = [\mathbf{z}_1, \cdots, \mathbf{z}_g \cdots, \mathbf{z}_c]$, the $\mathsf{Pr}(G) = [\mathsf{Pr}_1(G), \cdots, \mathsf{Pr}_g \cdots, \mathsf{Pr}_c(G)]$, because the $\sum_{i=1}^{|\mathcal{V}^0 \cup \mathcal{V}^1|} \sum_{l=1}^{|\Delta \mathcal{A}|} \Phi^l_i = \Delta \mathbf{z} = \Delta \mathbf{z}(G_1) - \Delta \mathbf{z}(G_0)$, the objective function for the graph classification task is:

$$
\mathbf{x}^* = \underset{\substack{\mathbf{x} \in \{0,1\}^{|\Delta \mathcal{A}|} \\ \|\mathbf{x}\|_1 = n}}{\arg \min} \sum_{g=1}^{c} \left( -\mathsf{Pr}_g(G_1) \sum_{i=1}^{|\mathcal{V}^0 \cup \mathcal{V}^1|} \sum_{l=1}^{|\Delta \mathcal{A}|} x_l \Phi^l_{i,g} \right)
$$
$$
+ \log \sum_{g'=1}^{c} \exp \left( z_{g'}(G_0) + \sum_{i=1}^{|\mathcal{V}^0 \cup \mathcal{V}^1|} \sum_{l=1}^{|\Delta \mathcal{A}|} x_l \Phi^l_{i,g'} \right) \tag{16}
$$

### A.5 SELECTING THE IMPORTANT LAYER EDGES FOR LINK PREDICTION

Selecting the important layer edges for link prediction task can be seen in Algorithm 2.

---

**Algorithm 2** Selecting important layer edges to explain evolution of $\mathsf{Pr}(Y|G_0)$ to $\mathsf{Pr}(Y|G_1)$ on the link prediction task

---

1: **Input**: the source graph $G_0$ and the destination graph $G_1$, Pre-trained GNN parameters $\boldsymbol{\theta}$
2: Obtain the layer edges flow set $\Delta\boldsymbol{\mathcal{A}}$
3: Initialize layer edges contribution matrix $\Phi \in \mathbb{R}^{|\Delta\boldsymbol{\mathcal{A}}| \times c}$ as an all-zero matrix
4: Obtain the altered massage flows set $\Delta\boldsymbol{\mathcal{F}}$
5: Given the target edge $IJ$, $\Delta\boldsymbol{\mathcal{F}} = \{\mathcal{F} : \mathcal{F} \in \Delta\boldsymbol{\mathcal{F}} \text{ and } (\mathcal{F}[T] = I \text{ or } \mathcal{F}[T] = J)\}$
6: **for** $s$ for 1 to $|\Delta\boldsymbol{\mathcal{F}}|$ **do**
7:     Select the $s$-th message flow in $|\Delta\boldsymbol{\mathcal{F}}|$ and calculate $\mathbf{C}_s$ according to the Eq. (13)
8:     Obtain the changed layer edges set $\Delta\boldsymbol{\mathcal{A}}_{\mathcal{F}}$ on this flow
9:     **for** $a^t_{\mathcal{F}[t-1]\mathcal{F}[t]}$ in $\Delta\boldsymbol{\mathcal{A}}_{\mathcal{F}}$ **do**
10:         According to the section 3.2, calculate $\phi_{a^t_{\mathcal{F}[t-1]\mathcal{F}[t]}}(\mathcal{F})$
11:         Let the index of $a^t_{\mathcal{F}[t-1]\mathcal{F}[t]}$ in $\Delta\boldsymbol{\mathcal{A}}$ is $l$, $\Phi_l = \Phi_l + \phi_{a^t_{\mathcal{F}[t-1]\mathcal{F}[t]}}(\mathcal{F})$
12:     **end for**
13: **end for**
14: Solve Eq. (15) to obtain the important changed layer edges
15: **Output**: The important changed layer edges set

---

### A.5.1 SELECTING THE IMPORTANT LAYER EDGES FOR GRAPH CLASSIFICATION

Selecting the important layer edges for graph classification task can be seen in Algorithm 3.

---

**Algorithm 3** Selecting important layer edges to explain evolution of $\mathsf{Pr}(Y|G_0)$ to $\mathsf{Pr}(Y|G_1)$ on the graph classification tasks

---

1: **Input**: the source graph $G_0$ and the destination graph $G_1$, Pre-trained GNN parameters $\boldsymbol{\theta}$
2: Obtain the layer edges flow set $\Delta\boldsymbol{\mathcal{A}}$
3: Initialize layer edges contribution matrix $\Phi^l \in \mathbb{R}^{|\mathcal{V}^0 \cup \mathcal{V}^1| \times c}$ as an all-zero matrix
4: **for** $s$ for 1 to $|\Delta\boldsymbol{\mathcal{F}}|$ **do**
5:     Select the $s$-th message flow in $|\Delta\boldsymbol{\mathcal{F}}|$ and calculate $\mathbf{C}_s$ according to the Eq. (14)
6:     obtain the changed layer edges set $\Delta\boldsymbol{\mathcal{A}}_{\mathcal{F}}$ on this flow
7:     **for** $a^t_{\mathcal{F}[t-1]\mathcal{F}[t]}$ in $\Delta\boldsymbol{\mathcal{A}}_{\mathcal{F}}$ **do**
8:         According to the section 3.2, calculate $\phi_{a^t_{\mathcal{F}[t-1]\mathcal{F}[t]}}(\mathcal{F})$
9:         Let the index of $a^t_{\mathcal{F}[t-1]\mathcal{F}[t]}$ in $\Delta\boldsymbol{\mathcal{A}}$ is $l$. Let the index of $\mathcal{F}[T]$ in the $\mathcal{V}^0 \cup \mathcal{V}^1$ is $i$
10:         $\Phi^l_i = \Phi^l_i + \phi_{a^t_{\mathcal{F}[t-1]\mathcal{F}[t]}}(\mathcal{F})$
11:     **end for**
12: **end for**
13: Solving the Eq. (16) to obtain the important changed layer edges
14: **Output**: The important changed layer edges set

---

## A.6 OBTAIN THE IMPORTANT INPUT EDGES

### A.6.1 ON THE NODE CLASSIFICATION TASK

Let $\Phi$ denotes the contribution matrix of edges, where $\Phi_l$ represents the contribution of $l$-th edge to $\Delta\mathbf{z}_J$, and $\Phi_{l,k}$ indicates the contribution of $l$-th edge to $\Delta z_k$, we can define the following objective function for the node classification:

$$
\begin{aligned}
\mathbf{x}^* &= \underset{\substack{\mathbf{x} \in \{0,1\}^{|\Delta\mathcal{E}|} \\ \|\mathbf{x}\|_1 = n}}{\arg\min} \sum_{k=1}^{c} \left( -\mathsf{Pr}_k(G_1) \sum_{l=1}^{|\Delta\mathcal{E}|} x_l \Phi_{l,k} \right) \\
&\quad + \log \sum_{k'=1}^{c} \exp \left( z_{k'}(G_0) + \sum_{l=1}^{|\Delta\mathcal{E}|} x_l \Phi_{l,k'} \right)
\end{aligned}
\tag{17}
$$

### A.6.2 ON THE LINK PREDICTION TASK

For the link prediction, the $\mathbf{z}_{IJ}(G) = [\mathbf{z}_1, \cdots, \mathbf{z}_\ell \cdots, \mathbf{z}_c], \mathsf{Pr}_{IJ}(G) = [\mathsf{Pr}_1(G), \cdots, \mathsf{Pr}_\ell \cdots, \mathsf{Pr}_c(G)]$, Let $\Phi$ denotes the contribution matrix of edges, where $\Phi_l$ represents the contribution of $l$-th edge to $\Delta \mathbf{z}_{IJ}$, and $\Phi_{l,\ell}$ indicates the contribution of $l$-th edge to $\Delta z_\ell$, we can define the following objective function for the link prediction:

$$\mathbf{x}^* = \underset{\substack{\mathbf{x} \in \{0,1\}^{|\Delta \mathcal{E}|} \\ \|\mathbf{x}\|_1 = n}}{\arg\min} \sum_{\ell=1}^{c} \left( -\mathsf{Pr}_\ell(G_1) \sum_{l=1}^{|\Delta \mathcal{E}|} x_l \Phi_{l,\ell} \right)$$

$$+ \log \sum_{\ell'=1}^{c} \exp \left( z_{\ell'}(G_0) + \sum_{l=1}^{|\Delta \mathcal{E}|} x_l \Phi_{l,\ell'} \right) \tag{18}$$

### A.6.3 ON THE GRAPH CLASSIFICATION TASK

For the graph classification, the $\Phi^l$ denotes contribution matrix of the $l$-th layer edge in the $\Delta \mathcal{A}$. The logits of the graph classification $\mathbf{z}_G = [\mathbf{z}_1, \cdots, \mathbf{z}_g \cdots, \mathbf{z}_c]$, the $\mathsf{Pr}(G) = [\mathsf{Pr}_1(G), \cdots, \mathsf{Pr}_g \cdots, \mathsf{Pr}_c(G)]$, because the $\sum_{i=1}^{|\mathcal{V}^0 \cup \mathcal{V}^1|} \sum_{l=1}^{|\Delta \mathcal{A}|} \Phi_i^l = \Delta \mathbf{z} = \Delta \mathbf{z}(G_1) - \Delta \mathbf{z}(G_0)$, the objective function for the graph classification task is:

$$\mathbf{x}^* = \underset{\substack{\mathbf{x} \in \{0,1\}^{|\Delta \mathcal{A}|} \\ \|\mathbf{x}\|_1 = n}}{\arg\min} \sum_{g=1}^{c} \left( -\mathsf{Pr}_g(G_1) \sum_{i=1}^{|\mathcal{V}^0 \cup \mathcal{V}^1|} \sum_{l=1}^{|\Delta \mathcal{E}|} x_l \Phi_{i,g}^l \right)$$

$$+ \log \sum_{g'=1}^{c} \exp \left( z_{g'}(G_0) + \sum_{i=1}^{|\mathcal{V}^0 \cup \mathcal{V}^1|} \sum_{l=1}^{|\Delta \mathcal{E}|} x_l \Phi_{i,g'}^l \right) \tag{19}$$

### A.6.4 SELECTING THE IMPORTANT INPUT EDGES

Selecting the important input edges for node classification and link prediction can be seen in the Algorithm 4. The selection of important input edges for graph classification can be seen in the Algorithm 5.

## A.7 EXPERIMENTS

### A.7.1 DATASETS

We study node classification task on the YelpChi, YelpNYC (Rayana & Akoglu, 2015), Pheme (Zubiaga et al., 2017) and Weibo (Ma et al., 2018) datasets. We explore the link prediction tasks on the BC-OTC, BC-Alpha, and UCI datasets. We study the graph classification tasks on MUTAG (Debnath et al., 1991), ClinTox, IMDB-BINARY and REDDIT-BINARY datasets. The details of data are in Table 2.

In the simulated dynamic graphs, we modify edge weights without adding or removing edges. Specifically, given a changed ratio $r$, we randomly adjust the the weights of $|\mathcal{E}^0| \times r$ edges to create evolving graphs. For the real dynamic graph datasets used in the node classification and link prediction tasks, timestamps allow us to track graph evolution, which includes modifications to edge weights, as well as the addition and deletion of edges. In graph classification, we apply slight perturbations to the graphs (You et al., 2018), by randomly adding or removing edges or altering edge weights.

- YelpChi, YelpNYC (Rayana & Akoglu, 2015): each node represents a review, product, or user. If a user posts a review to a product, there are edges between the user and the review, and between the review and the product. The data sets are used for node classification.

- Pheme (Zubiaga et al., 2017) and Weibo (Ma et al., 2018): they are collected from Twitter and Weibo. A social event is represented as a trace of information propagation. Each event has a label, rumor or non-rumor. Consider the propagation tree of each event as a graph. The data sets are used for node classification.

---

**Algorithm 4** Selecting important input edges to explain evolution of $\Pr(Y|G_0)$ to $\Pr(Y|G_1)$ on the node classification and link prediction tasks

---

1: **Input**: the source graph $G_0$ and the destination graph $G_1$, Pre-trained GNN parameters $\boldsymbol{\theta}$
2: Obtain the changed edges set $\Delta\mathcal{E} = \{a_{UV} : a_{UV}^0 \neq a_{UV}^1, t \in \{1, \dots, T\}, U, V \in \mathcal{V}^0 \cup \mathcal{V}^1\}$
3: Initialize layer edges contribution matrix $\Phi \in \mathbb{R}^{|\Delta\mathcal{E}| \times c}$ as an all-zero matrix
4: Obtain the altered massage flows set $\Delta\mathcal{F} = \{\mathcal{F} : \mathcal{F} = (\mathcal{F}[0], \dots, \mathcal{F}[t] \dots \mathcal{F}[T]), a_{\mathcal{F}[t-1]\mathcal{F}[t]}^{0,t} \neq a_{\mathcal{F}[t-1]\mathcal{F}[t]}^{1,t}, t = 1, \dots, T\}$
5: **if** The node classification task **then**
6:     Given the target node $J$, $\Delta\mathcal{F} = \{\mathcal{F} : \mathcal{F} \in \Delta\mathcal{F}$ and $\mathcal{F}[T] = J\}$
7: **else if** The link prediction task **then**
8:     Given the target edge $IJ$, $\Delta\mathcal{F} = \{\mathcal{F} : \mathcal{F} \in \Delta\mathcal{F}$ and $(\mathcal{F}[T] = I$ or $\mathcal{F}[T] = J)\}$
9: **end if**
10: **for** $\mathcal{F}$ in $|\Delta\mathcal{F}|$ **do**
11:     According to the Eq. (7) (node classification) or Eq. (13) (link prediction), calculate the message flow contribution $\mathbf{c}$
12:     obtain the changed edges set $\Delta\mathcal{E}_{\mathcal{F}} = \{a_{\mathcal{F}[t-1]\mathcal{F}[t]} : a_{\mathcal{F}[t-1]\mathcal{F}[t]}^0 \neq a_{\mathcal{F}[t-1]\mathcal{F}[t]}^1\}$ on this flow
13:     **for** $a_{\mathcal{F}[t-1]\mathcal{F}[t]}$ in $\Delta\mathcal{E}_{\mathcal{F}}$ **do**
14:         According to the Section 3.2, calculate $\phi_{a_{\mathcal{F}[t-1]\mathcal{F}[t]}}(\mathcal{F})$.
15:         Let the $a_{\mathcal{F}[t-1]\mathcal{F}[t]}$ is the $l$-th edge in $\Delta\mathcal{E}$, $\Phi_l = \Phi_l + \phi_{a_{\mathcal{F}[t-1]\mathcal{F}[t]}}(\mathcal{F})$
16:     **end for**
17: **end for**
18: Solving the Eq. (17) (node classification) or Eq. (18) (link prediction) to obtain the important changed input edges
19: **Output**: The important changed input edges set

---

---

**Algorithm 5** Selecting the important input edges to explain evolution of $\Pr(Y|G_0)$ to $\Pr(Y|G_1)$ on the graph classification tasks

---

1: **Input**: the source graph $G_0$ and the destination graph $G_1$, Pre-trained GNN parameters $\boldsymbol{\theta}$
2: Obtain the layer edges flow set $\Delta\mathcal{A} = \{a_{UV}^t : a_{UV}^{0,t} \neq a_{UV}^{1,t}, t \in \{1, \dots, T\}, U, V \in \mathcal{V}^0 \cup \mathcal{V}^1\}$
3: Obtain the altered massage flows set $\Delta\mathcal{F} = \{\mathcal{F} : \mathcal{F} = (\mathcal{F}[0], \dots, \mathcal{F}[t] \dots \mathcal{F}[T]), a_{\mathcal{F}[t-1]\mathcal{F}[t]}^{0,t} \neq a_{\mathcal{F}[t-1]\mathcal{F}[t]}^{1,t}, t = 1, \dots, T\}$
4: **for** $l$ for 1 to $|\Delta\mathcal{A}|$ **do**
5:     Initialize layer edges contribution matrix $\Phi^l \in \mathbb{R}^{|\mathcal{V}^0 \cup \mathcal{V}^1| \times c}$ as an all-zero matrix
6: **end for**
7: **for** $\mathcal{F}$ in $|\Delta\mathcal{F}|$ **do**
8:     According to the Eq. (14), calculate the message flow contribution $\mathbf{c}$
9:     obtain the changed edges set $\Delta\mathcal{E}_{\mathcal{F}} = \{a_{\mathcal{F}[t-1]\mathcal{F}[t]} : a_{\mathcal{F}[t-1]\mathcal{F}[t]}^0 \neq a_{\mathcal{F}[t-1]\mathcal{F}[t]}^1\}$ on this flow
10:     **for** $a_{\mathcal{F}[t-1]\mathcal{F}[t]}^t$ in $\Delta\mathcal{E}_{\mathcal{F}}$ **do**
11:         According to the section 3.2, calculate $\phi_{a_{\mathcal{F}[t-1]\mathcal{F}[t]}^t}(\mathcal{F})$
12:         Let the $a_{\mathcal{F}[t-1]\mathcal{F}[t]}^t$ is the $l$-th layer edge in $\Delta\mathcal{E}$, $\mathcal{F}[T]$ is the $i$-th node in the $\mathcal{V}^0 \cup \mathcal{V}^1$, $\Phi_i^l = \Phi_i^l + \phi_{a_{\mathcal{F}[t-1]\mathcal{F}[t]}^t}(\mathcal{F})$
13:     **end for**
14: **end for**
15: Solving the Eq. (19) to obtain the important changed input edges
16: **Output**: The important changed input edges set

---

- BC-OTC[1] and BC-Alpha[2]: is a who trusts-whom network of bitcoin users trading on the platform. The data sets are used for link prediction.

- UCI[3]: is an online community of students from the University of California, Irvine, where in the links of this social network indicate sent messages between users. The data sets are used for link prediction.

- MUTAG (Debnath et al., 1991): A molecule is represented as a graph of atoms where an edge represents two bounding atoms.

- ClinTox (Gayvert et al., 2016):compares drugs approved through FDA and drugs eliminated due to the toxicity during clinical trials.

- IMDB-BINARY is movie collaboration datasets. Each graph corresponds to an ego-network for each actor/actress, where nodes correspond to actors/actresses and an edge is drawn betwen two actors/actresses if they appear in the same movie.Each graph is derived from a pre-specified genre of movies, and the task is to classify the genre graph it is derived from.

- REDDIT-BINARY is balanced datasets where each graph corresponds to an online discussion thread and nodes correspond to users. An edge was drawn between two nodes if at least one of them responded to another's comment. The task is to classify each graph to a community or a subreddit it belongs to.

Table 2: The details of datasets

| Datasets | Nodes(Avg. Nodes) | Edges(Avg. Edges) | task | Accuracy(AUC) |
|---|---|---|---|---|
| YelpChi | 105,659 | 375,239 | node classification | 0.8477 |
| YelpNYC | 520,200 | 1,956,408 | node classification | 0.8743 |
| weibo | 4,657 | - | node classification | 0.9549 |
| pheme | 5,748 | - | node classification | 0.7621 |
| BC-OTC | 5,881 | 35,588 | link prediction | 0.9388 |
| BC-Alpha | 3,777 | 24,173 | link prediction | 0.9125 |
| UCI | 1,899 | 59,835 | link prediction | 0.9061 |
| MUTAG | 17.93 | 19.79 | graph classification | 0.75 |
| ClinTox | 26.1 | 55.5 | graph classification | 0.9874 |
| IMDB-BINARY | 19.8 | 193.1 | graph classification | 0.8 |
| REDDIT-BINARY | 429.6 | 995.5 | graph classification | 0.716 |

Table 3: The changed ration $r$ on different datasets

| YelpChi | YelpNYC | Weibo | Pheme | BC-OTC | BC-Alpha | UCI | MUTAG | ClinTox | IMDB-BINARY | REDDIT-BINARY |
|---|---|---|---|---|---|---|---|---|---|---|
| 1 | 1 | 1 | 1 | 0.5 | 0.6 | 0.4 | 1 | 1 | 1 | 1 |

### A.7.2 BASSLINES

- **GNNExplainer** is designed to explain GNN predictions for node and graph classification on static graphs. We train the explainer on graphs $G_0$ and $G_1$ to obtain the edges contribution $\Phi^0$ and $\Phi^1$. The final edges contribution is given by $\Phi = \Phi^1 - \Phi^0$ if the predicted class on $G_0$ and $G_1$ are different. Otherwise, $\Phi = \Phi^1$. The top-K edges are selected based on $\Phi$ as the explanations.

- **PGExplainer** learns approximated discrete masks for edges to explain the predictions, with important edges selected in the same manner as GNNExplainer.

- **GNN-LRP** utilizes the back-propagation attribution method LRP to GNN (Schnake et al., 2020), attributing the class probability $\Pr(Y = k|G_1)$ to input neurons regardless of $\Pr(Y|G_0)$, thereby obtaining contribution scores for message flows. It uses a summation function to map these contributions to edges, with edge selection consistent with GNNExplainer.

---

[1] http://snap.stanford.edu/data/soc-sign-bitcoin-otc.html
[2] http://snap.stanford.edu/data/soc-sign-bitcoin-alpha.html
[3] http://konect.cc/networks/opsahl-ucsocial

- **DeepLIFT** (Shrikumar et al., 2017) attributes the log-odd between two probabilities $\Pr(Y = k|G_0)$ and $\Pr(Y = k'|G_1)$, where $k \neq k'$, to the message flows. Then it uses a summation function to obtain contributions of edges. The edge selection process is consistent with GNNExplainer.

- **FlowX** applies the Shapley value to derive initial contributions of message flows, subsequently training these scores by defining loss functions. A summation function is employed to map contributions to edges, with edge selection aligned with GNNExplainer.

- **AxiomLayeredge-Topk** is a variant of AxiomLayeredge that selects the top layer edges based on the highest contributions $\Phi \mathbf{1}$, where $\mathbf{1}$ is an all-1 $c \times 1$ vector.

- **AxiomEdge** maps the contribution of message flows to the input edges also using the Shapley value. See Algorithm 4 and Algorithm 5 in the Appendix A.6 for details.

- **AxiomEdge-Topk** is a variant of AxiomEdge that selects the top edges with the highest contributions $\Phi_{\mathcal{E}} \mathbf{1}$, where $\Phi_{\mathcal{E}}$ is the contribution matrix of the altered edges, $\mathbf{1}$ is an all-1 $c \times 1$ vector.

- **AxiomEdge\Shapley** is a variant of AxiomEdge that utilizes the average function instead of the Shapley value when mapping contributions of message flows to edges.

- **AxiomLayeredge\Shapley** is a variant of AxiomLayeredge that utilizes the average function instead of the Shapley value when mapping the contribution of message flow to layer edges.

### A.7.3 EXPERIMENTAL SETUP

We trained the two layers GNN. utilizing element-wise sum as the aggregation function $f_{AGG}$. The logit for node $J$ is denoted by $z_J(G)$. For node classification, $z_J(G)$ is mapped to the class distribution through the softmax function. For the link prediction, we concatenate $z_I(G)$ and $z_J(G)$ as the input to a linear layer to obtain the logits, which are then mapped to the probability of the existence of the edge $(I, J)$. For the graph classification task, the average pooling of $z_J(G)$ across all nodes in $G$ can produce a single vector representation $z(G)$ for classification. It can be mapped to the class probability distribution through the softmax function. During training, we set the learning rate to 0.01, the dropout rate to 0.2 and the hidden size to 16. The model is trained and then fixed during the prediction and explanation stages.

### A.7.4 THE PREDEFINED SPARSITY

On the real dynamic graphs, the sparsity of explanations across various datasets and tasks is illustrated in Table 4. The sparsity of simulated dynamic graphs is illustrated in Table 5. The sparsity is small, but our method can also achieve the better performance than the baselines.

Table 4: The sparsity of explanations on real dynamic graph datasets

| Datasets | Sparsity level 1 | Sparsity level 2 | Sparsity level 3 | Sparsity level 4 | Sparsity level 5 |
|---|---|---|---|---|---|
| **YelpChi** | 0.996 | 0.992 | 0.988 | 0.994 | 0.98 |
| **YelpNYC** | 0.998 | 0.997 | 0.996 | 0.995 | 0.994 |
| **weibo** | 0.996 | 0.993 | 0.99 | 0.986 | 0.982 |
| **pheme** | 0.98 | 0.96 | 0.94 | 0.92 | 0.9 |
| **BC-OTC** | 0.996 | 0.995 | 0.994 | 0.993 | 0.992 |
| **BC-Alpha** | 0.995 | 0.994 | 0.993 | 0.992 | 0.991 |
| **UCI** | 0.998 | 0.997 | 0.996 | 0.994 | 0.992 |
| **MUTAG** | 0.988 | 0.976 | 0.964 | 0.952 | 0.94 |
| **ClinTox** | 0.991 | 0.982 | 0.973 | 0.964 | 0.954 |
| **IMDB-BINARY** | 0.996 | 0.991 | 0.988 | 0.984 | 0.98 |
| **REDDIT-BINARY** | 0.998 | 0.997 | 0.996 | 0.995 | 0.994 |

### A.7.5 EVALUATION METRIC

In Figure 7, we illustrate the calculation process of evaluation metric.

### A.7.6 PERFORMANCE EVALUATION AND COMPARISON

We compare the performance of the methods across three tasks: node classification, link prediction and graph classification in simulate dynamic graph scene, as illustrated in Figure 8. For

Table 5: The sparsity of explanations on different simulated graph datasets

| Datasets | Sparsity level 1 | Sparsity level 2 | Sparsity level 3 | Sparsity level 4 | Sparsity level 5 |
|---|---|---|---|---|---|
| YelpChi | 0.999 | 0.998 | 0.997 | 0.996 | 0.995 |
| YelpNYC | 0.9994 | 0.9988 | 0.9981 | 0.9975 | 0.9965 |
| weibo | 0.9972 | 0.9945 | 0.992 | 0.989 | 0.986 |
| pheme | 0.982 | 0.963 | 0.945 | 0.927 | 0.908 |
| BC-OTC | 0.967 | 0.95 | 0.935 | 0.918 | 0.9 |
| BC-Alpha | 0.95 | 0.91 | 0.87 | 0.83 | 0.79 |
| UCI | 0.999 | 0.998 | 0.997 | 0.996 | 0.995 |
| MUTAG | 0.988 | 0.976 | 0.964 | 0.952 | 0.94 |
| ClinTox | 0.99 | 0.98 | 0.97 | 0.96 | 0.95 |
| IMDB-BINARY | 0.996 | 0.992 | 0.988 | 0.984 | 0.98 |
| REDDIT-BINARY | 0.998 | 0.996 | 0.994 | 0.992 | 0.99 |

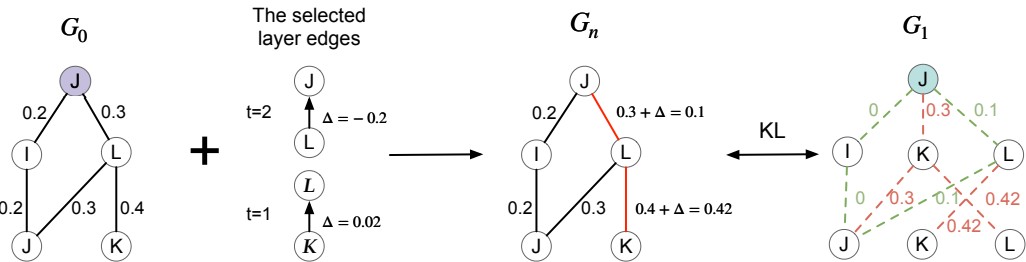

Figure 7: The calculation process of evaluation metric.

each dataset, we report the average KL over target nodes/edges/graphs. From Figure 8, we can see that our method AxiomLayeredge has the smallest KL across all levels of explanation sparsity and datasets and tasks, with exception of Weibo, Pheme and certain sparsity levels of YelpNYC dataset. In datasets with dense graph structures (YelpChi, YelpNYC, BC-Alpha, BC-OTC, UCI, IMDB-BINARYdand REDDIT-BINAYR), the AxiomLayeredge-TopK method ranks third. This indicates that our designed message flow contribution value Algorithm can effectively explain the dynamic graphs. In seven experimental settings (Weibo, YelpChi, YelpNYC, BC-Alpha, UCI, MUTAG, ClinTox), our method AxiomLayeredge along with its variants AxiomEdge, AxiomEdge\Shapley, AxiomLayeredge\Shapley outperform the GNNLRP, DeepLIFT, GNNExplainer, PGExplainer and FlowX methods. This demonstrates that our proposed methods more effectively explain the evolution of $\Pr(Y|G_0; \theta)$ to $\Pr(Y|G_1; \theta)$, while methods designed for static graph struggle to identify salient edges that explain changes in the predicted probability distribution.

### A.7.7 RUNNING TIME

We plot the running time for searching $\Delta \mathcal{F}$, calculating message flow contributions, using the Shapley value to attribute contributions to layer edges, and selecting important layer edges on the Pubmed, Coauthor-Computer, and Coauthor-Physics datasets. The details of these datasets are shown in the Table 6. As shown in Figure 9, the larger $\Delta \mathcal{A}$ lead to higher cost in the selecting step compared to the other steps. The time for calculating contributions and applying the Shapley value remains relatively small, even for larger graphs. On large graphs, the searching and selecting steps dominate the running time, but the overall time remains manageable. In practice, incremental message flow searches tailored to specific graph topologies and more efficient optimization algorithms can further speed up the process.

- In citation network, PubMed (Kipf & Welling, 2017), each paper has bag-of-words features, and the goal is to predict the research area of each paper.

- Coauthor-Computer and Coauthor-Physics are co-authorship graphs based on the Microsoft Academic Graph from the KDD Cup 2016. We represent authors as nodes, that are connected by an edge if they co-authored a paper (Shchur et al., 2018). Node features represent paper keywords for each author's papers.

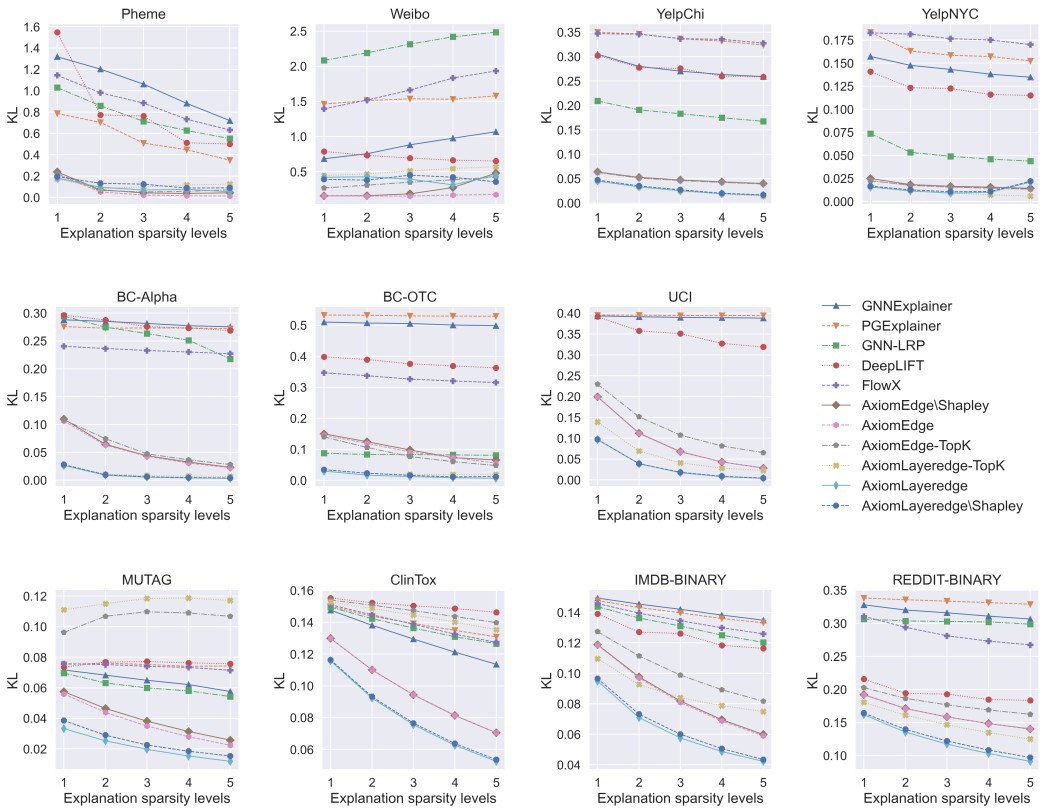

Figure 8: Performance in KL as $G_0 \rightarrow G_1$. Each column corresponds to a different dataset. The first, second and third rows represent node classification, link prediction and graph classification tasks, respectively.

Table 6: Three large graph datasets

| Datasets | Classes | Nodes | Edges | Edge/Node | Features |
|---|---|---|---|---|---|
| **PubMed** | 3 | 19,717 | 44,324 | 2.24 | 500 |
| **Coauthor-Computer** | 13 | 18,333 | 327,576 | 17.87 | 6,805 |
| **Coauthor-Physics** | 2 | 34,493 | 991,848 | 28.76 | 8,415 |

### A.7.8 VISUALIZATION AND ACCURACY ON THE BA-SHAPES DATESET

On the BA-Shapes dataset, we randomly generated 1,000 graphs with a House motif and 1,000 with a Circle motif. For each motif dataset, we randomly deleted one edge to disrupt the motif and perturbed edges outside the motif area, generating another 1,000 graph datasets. We trained a GNN model to classify the presence of the motif. We applied explanation methods to select one edge. If the selected edge disrupts the motif, the explanation is correct, while if the edge lies outside the motif area, the explanation is wrong. The accuracy results for GNN and the explanation methods are presented in Table 8 and Table 7, respectively. Visualization of the explanations for the House and Circle motifs are shown in Figure 10 and Figure 11.

Table 7: The accuracy of explanation methods

| Datasets | our | GNNExplainer | PGExplainer | DeepLIFT | GNN-LRP | FlowX |
|---|---|---|---|---|---|---|
| **Circle-motif** | 0.9657 | 0.9067 | 0.4765 | 0.8848 | 0.1152 | 0.4156 |
| **House-motif** | 0.9936 | 0.3618 | 0.6025 | 0.9897 | 0.1755 | 0.0238 |

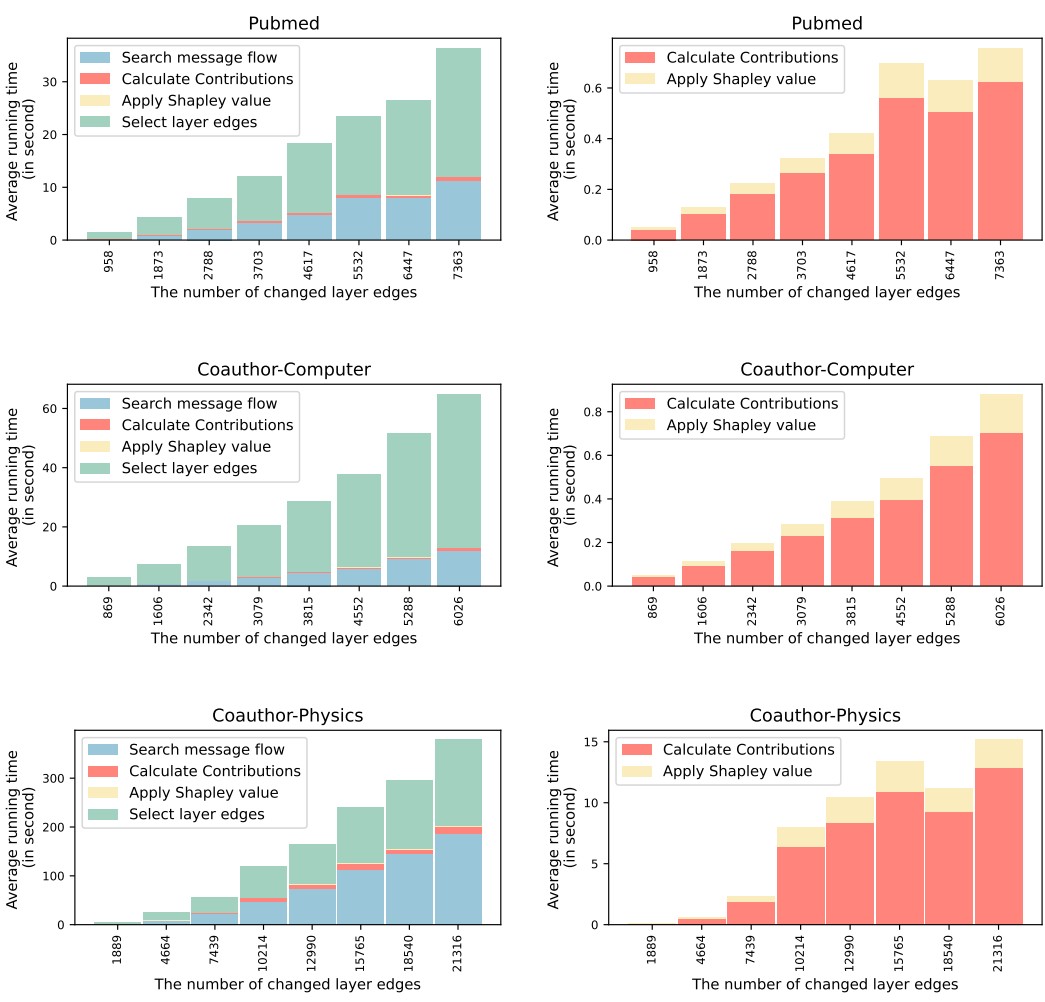

Figure 9: Running time decomposition: Each row represents a dataset. The first column shows the total running time for all four steps, while the second column displays the running time for calculating contributions and applying the Shapley value.

Table 8: The accuracy of GNN model

| Datasets | GCN |
|---|---|
| **Circle-motif** | 0.755 |
| **House-motif** | 0.847 |

### A.7.9 PERFORMANCE EVALUATION ON DISCONTINUOUS CHANGES OF EDGES

We evaluate the effectiveness of our method on node classification, link prediction, and graph classification tasks under discontinuous edge changes. As shown in Figure 12, our method outperforms others across all five datasets, further validating its effectiveness.

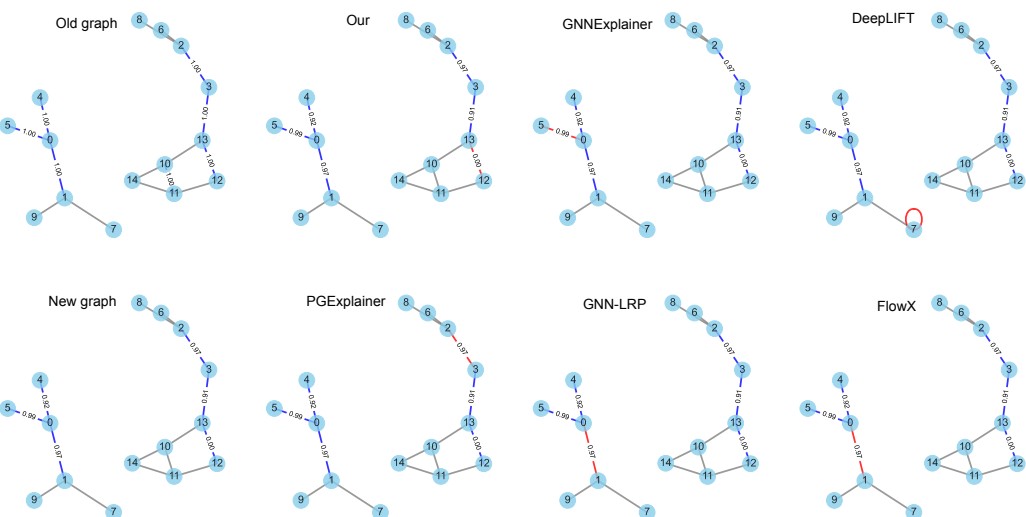

Figure 10: The visualization of House-motif dataset. The blue edges represent changes in edge weights. In the new graph, the edge (12, 13) is removed to destroy the motif, and the weights of edges (0, 1), (2,3), (0, 4), (13, 3), and (0, 5) are perturbed. The edge (12, 13) serves as the ground truth for the explanation, clarifying why the old graph contains a house while the new graph does not. The red edge represents the selected edge by different methods. Our method correctly identifies (12, 13) as the explanation, while other methods select the wrong edge.

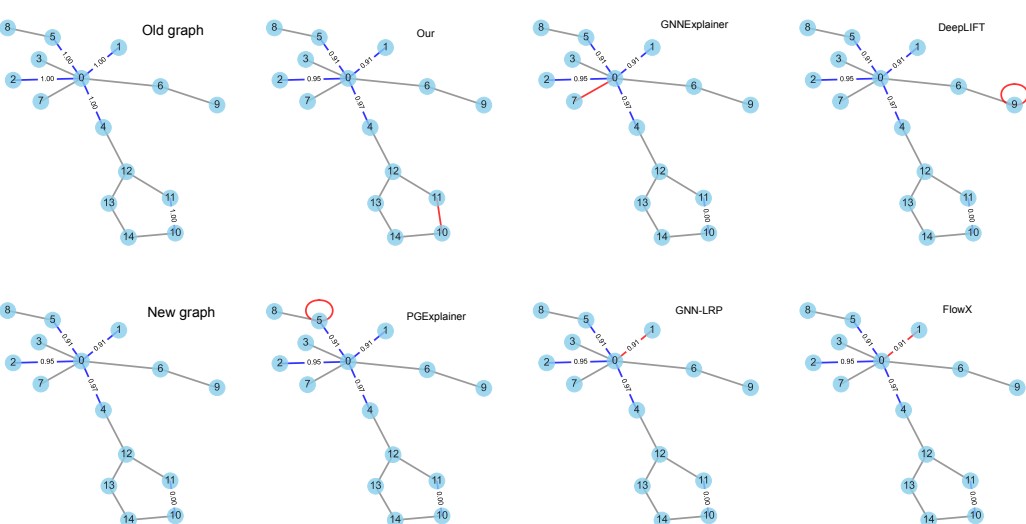

Figure 11: The visualization of Circle-motif dataset. The blue edges represent changes in edge weights. In the new graph, the edge (10, 11) is removed to destroy the motif, and the weights of edges (0, 1), (0, 4), (0, 2), and (0, 5) are perturbed. The edge (10, 11) serves as the ground truth for the explanation, clarifying why the old graph contains a circle while the new graph does not. The red edge represents the selected edge by different methods. Our method correctly identifies (10, 11) as the explanation, while other methods select the wrong edge.

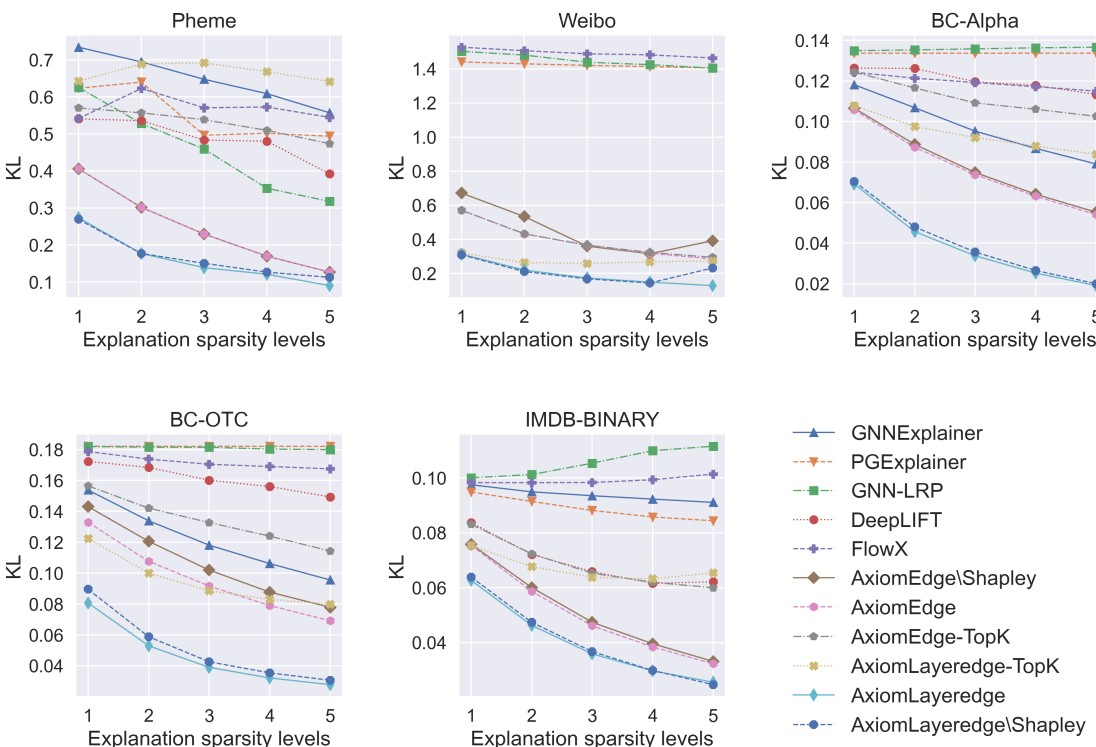

Figure 12: Performance in KL as $G_0 \rightarrow G_1$ when only adding or deleting edges. Each column corresponds to a different dataset.

