# OpenReview forum: "Explanations of GNN on Evolving Graphs via Axiomatic  Layer edges"
_ICLR.cc/2025/Conference — ICLR 2025 Poster_

### Official Review · Reviewer_KvnM · 2024-11-02

**Soundness:** 3
**Presentation:** 3
**Contribution:** 4
**Rating:** 6
**Confidence:** 4

**Summary:**

This paper provides explanations for predictions on evolving graphs. In the case of continuously changing edge weights, the authors propose an explanation method based on Axiomatic Layer Edges, mapping the contribution of message flows to layer edges using Shapley values, and selecting key layer edges by minimizing the KL divergence optimization problem. Experimental results show that this method outperforms existing baseline methods across multiple datasets and tasks.

**Strengths:**

1. The paper presents an explanation method based on layer edges, which can more accurately capture the variations in message passing, thereby improving the fidelity of the explanations. The method is innovative.
2. The authors explain prediction changes by selecting a small number of key layer edges at multiple scales through a KL divergence-based optimization problem.
3. The paper conducts experiments on multiple datasets and tasks, including node classification, link prediction, and graph classification tasks.

**Weaknesses:**

1. The authors should conduct a complexity analysis, as the computation of Shapley values generally has a high complexity. Additionally, since the authors select a small number of key layer edges in their method, will this impact the complexity of the model?
2. In this paper, the authors use a series of carefully designed methods to balance the fidelity and interpretability of explanations. However, I still have a question: is there an optimal balance point (for example, the optimal point reached in Figure 2(b))?
3. The paper assumes that changes in edge weights are continuous; however, could the authors also explore sudden or discontinuous changes (such as the addition or deletion of nodes or edges)? Changes in graph structure can be quite complex and unpredictable, which may affect the robustness of the method.

**Questions:**

See Weaknesses.

---

> ### Author Response · Authors · 2024-11-20
> **Responses to Reviewer KvnM**
>
> Thank you for taking the time to review our paper! Kindly note that we have uploaded a **revised version**, which includes additional details for your review.
>
> > The authors should conduct a complexity analysis, as the computation of Shapley values generally has a high complexity. Additionally, since the authors select a small number of key layer edges in their method, will this impact the complexity of the model?
>
> In **Section 3.4**, we provide a complexity analysis. The computational complexity of calculating the Shapley value is $O\big(|\Delta \mathcal{F}| (2^{T}-1) d^1 \cdots d^{T+1} \big)$, where $d^t$ and $d^{t+1}$ denote the dimension of the $\theta^{t} \in \mathbb{R}^{d^{t} \times d^{t+1}}, t=\{1,\cdots, T\}$. The most time-consuming steps of our algorithm are obtaining the changed message flows and selecting important layer edges, with complexities of  $O\big(|\Delta \mathcal{E}|^T \big)$ and $O\big(|\Delta \mathcal{A}|^3 \big)$, respectively. These complexities grow exponentially with the number of changed edges or layer edges. In Figure 9, we present the running time for large datasets. The results show that attributing contributions using the Shapley value is relatively small, even for larger graphs.
>
> The small number of key layer edges does not significantly affect the method's complexity. While it impacts the constraints, the shape of the objective function, and the optimization process, it typically does not alter the time complexity of the solution algorithm. The core time complexity of the solver is $O\big(|\Delta \mathcal{A}|^3 \big)$, and in most cases, the solver's time complexity dominates the overall solution time.
>
> > In this paper, the authors use a series of carefully designed methods to balance the fidelity and interpretability of explanations. However, I still have a question: is there an optimal balance point (for example, the optimal point reached in Figure 2(b))?
>
> For smaller graph datasets, we can achieve an optimal balance point. For example, **Figures 10 and 11** visualize explanations on the House-motif and Circle-motif datasets, with detailed experimental setup provided in **Appendix A.7.8**. The key edges identified by our method not only faithfully explain why the motif exists in the old graph but not in the new graph, but also enhance user interpretability. However, for larger datasets, finding the optimal balance point is more challenging. As shown in **Figure 9**, the number of changed layer edges in large graph datasets can exceed 20,000. In such cases, the optimal balance point can be approached by gradually increasing the number of selected key layer edges.
>
> > The paper assumes that changes in edge weights are continuous; however, could the authors also explore sudden or discontinuous changes (such as the addition or deletion of nodes or edges)? Changes in graph structure can be quite complex and unpredictable, which may affect the robustness of the method.
>
> We conducted experiments to evaluate the effectiveness of our method on node classification, link prediction, and graph classification tasks under discontinuous edge changes. The results, shown in **Figure 12**, demonstrate that our method outperforms all baseline approaches across five datasets, further validating its effectiveness.

---

> > ### Comment · Reviewer_KvnM · 2024-11-23
> > **Thanks**
> >
> > Thanks for your effort in the rebuttal. I will keep my score.

---

> > > ### Author Response · Authors · 2024-11-24
> > > **Thanks to Reviewer**
> > >
> > > Thank you for  taking the time to review our rebuttal. We sincerely appreciate your thoughtful evaluation of our work.

---

### Official Review · Reviewer_aPrr · 2024-11-04

**Soundness:** 3
**Presentation:** 3
**Contribution:** 3
**Rating:** 6
**Confidence:** 4

**Summary:**

This paper introduces a method for the explainability of evolving graphs, where edge weights of a graph changes over time. The authors explore this problem across three tasks of node classification, link prediction, and graph classification and show that their approach demonstrates superior performance compared to other baselines. Furthermore, in their problem setting Pr(Y |G0; θ) will evolve to Pr(Y |G1; θ), and they aim to explain the evolution of Pr(Y |G; θ) with respect to G0 → G1. They formulate the changes in hidden vectors on G0 and G1 and apply chain rule of DeepLIFT to assign the changes in logits to message flows. Finally, they show that their approach outperforms other baselines on KL divergence metric.

**Strengths:**

- The problem of explainability for Graph Neural Networks (GNNs) in the context of evolving graphs is both interesting and novel.

- The proposed model demonstrates superior performance compared to other baseline methods.

**Weaknesses:**

- The metric used for evaluation is not clearly justified. There are standard metrics, such as fidelity, that could provide a better understanding of the method's performance.
- For an explainability method, it is crucial to include at least one visualization of a dataset. This would allow readers to visually assess the method's effectiveness.
- It would be beneficial to experimentally demonstrate that this method works with different Graph Neural Networks (GNNs). Additionally, including a table that shows the performance of the pre-trained GNNs across various datasets would be helpful.

**Questions:**

- What pre-trained GNN has been used in this work?
- Could you provide visualizations of your results for datasets such as MUTAG?
- What is the rational behind choosing KL divergence as the evaluation metric?

---

> ### Author Response · Authors · 2024-11-20
> **Responses to Reviewer aPrr**
>
> Thank you for taking the time to review our paper! Kindly note that we have uploaded a **revised version**, which includes additional details for your review.
>
> > The metric used for evaluation is not clearly justified. There are standard metrics, such as fidelity, that could provide a better understanding of the method's performance. What is the rational behind choosing KL divergence as the evaluation metric?
>
> We apologize for the lack of clarity regarding the evaluation metric in the original submission. The metric used for the node classification task is the $KL(Pr_J(G_1)||Pr_J(G_n))$. The idea of this metric is similar to Fidelity-[1]. We obtain G_n by only "keeping" the important changed layer edges. Starting from the computational graph of $G_0$ , we adjust the weights of the selected layer edges to those in $G_1$ , while leaving the weights of unselected layer edges unchanged. An example of this process is shown in **Figure 7** in the Appendix. Intuitively, if only adjusting the selected layer edges brings $Pr_J(G_n)$ closer to $Pr_J(G_1)$, it indicates that these layer edges effectively explain the evolution from $Pr_J(G_0)$ to $Pr_J(G_1)$, leading to a smaller metric value. For the Fidelity-, it measures the diffenence of predicted probability for a specific class.
> However, we use KL divergence. The reason is that we want to explain the evolution of prediction probability distribution, not the probability for a given class. Besides, the the evolution of prediction probability distribution can provide a more comprehensive understanding of the model’s performance.
>
> > For an explainability method, it is crucial to include at least one visualization of a dataset. This would allow readers to visually assess the method's effectiveness. Could you provide visualizations of your results for datasets such as MUTAG?
>
> In **Appendix A.7.8**, we show experiments on the BA-Shapes dataset, where we randomly generate motif and non-motif datasets. We train a GNN model to classify motif presence and apply various explanation methods to the model. The **Figure 10** and **Figure 11** show the visualization of these dataset. These figures show that compared with different explanation methods, our method can always find the ground truth edge for explanations. The **Table 7** shows the accuracy of explanation methods on these two datasets. Our method has the highest accuracy for explanations.
>
> > It would be beneficial to experimentally demonstrate that this method works with different Graph Neural Networks (GNNs). Additionally, including a table that shows the performance of the pre-trained GNNs across various datasets would be helpful. What pre-trained GNN has been used in this work?
>
> The pre-trained GNN used in the paper is the GCN model. In **Table 2**, we show the performance of the pre-trained GNNs across various datasets. For the node classification and graph classification tasks, we show the accuracy. For the link prediction task, we show the AUC.
>
> For the different Graph Neural Networks, our methods can also be used in the GIN model.
>
> When our method is extended to the GAT model, we can only attribute changes in logits to changes in the attention weights of edges. Then we can select the small number of changed attention weights of edges as the explanations. Some work regard the attention weight as the explanations[2][3]. Thus the changed attention weights can also be considered as the explanation to explain the evolution of prediction probability distribution.
>
> But we cannot explain which added or removed edges contribute more or less to the changed attention weights, and how these added or removed edges impacts the evolution of prediction probability distribution. The reason is that we do not define the relus to attribute the difference of softmax function to the changed input. For example, we consider a target node J with four neighbors I,K,L,M. The attention for edge J--I is $\frac{e^a}{e^a+e^b+e^c+e^d}$, if we remove the edge J--L and the edge J--M, the attention for the IJ now is $\frac{e^a}{e^a+e^b}$. The problem is that we do not have the rules to attribute the $\frac{e^a}{e^a+e^b+e^c+e^d}-\frac{e^a}{e^a+e^b}$ to the changed input c and d. Thus, designing this rule is our future work. After obtaining this rule, we can explain the GAT model.
>
> **Reference**
>
> [1] Yuan H, Yu H, Gui S, et al. Explainability in graph neural networks: A taxonomic survey[J]. IEEE transactions on pattern analysis and machine intelligence, 2022, 45(5): 5782-5799.
>
> [2] Danish Pruthi, Mansi Gupta, Bhuwan Dhingra, Graham Neubig, and Zachary C. Lipton. 2020. Learning to deceive with attention-based explanations. In Proceedings of the 58th Annual Meeting of the Association for Computational Linguistics, pages 4782– 4793, Online. Association for Computational Linguistics.
>
> [3] Shikhar Vashishth, Shyam Upadhyay, Gaurav Singh Tomar, and Manaal Faruqui. 2019. Attention interpretability across NLP tasks. arXiv preprint arXiv:1909.11218.

---

> ### Comment · Reviewer_aPrr · 2024-11-23
>
> Thank you for your response. It would be beneficial to add the discussion of GAT and different Graph Neural Networks as a limitation in the main manuscript.

---

> > ### Author Response · Authors · 2024-11-24
> > **Response to the Reviewer’s Suggestion**
> >
> > Thank you for your valuable suggestion. Following your advice, we have **updated** our manuscript and added the discussion of GAT and different Graph Neural Networks as a limitation in **Section 5**: Limitations and Concerns.

---

### Official Review · Reviewer_7mCd · 2024-11-04

**Soundness:** 3
**Presentation:** 3
**Contribution:** 2
**Rating:** 6
**Confidence:** 2

**Summary:**

This paper works on the problem of explaining challenges in GNNs' predictions when the weights of the input edges are continuously changing. Prior work has weaknesses in axiomatic attribution of message flows, unfair distribution, and optimality. The proposed algorithm from the authors addresses the challenges by decomposing the changes in the message flow and introducing the Shapley value for fair attribution. Extensive experiments have been conducted to show the effectiveness of the proposed method. The paper is well-written and organized.

**Strengths:**

The intuition behind this paper is interesting and novel. The design of the framework is feasible and detailed explained. There are also sufficient experiments to demonstrate the effectiveness of the proposed methods.

**Weaknesses:**

The goal of this paper is to address the challenges of explanation on GNN in evolving graphs. Thus, it should include more dynamic graph baselines instead of simply using static GNNs. Besides, the authors should introduce more on the relationship between the proposed evaluator through the temporal dimension.

**Questions:**

1. There are typos in line 221. Please carefully check them.
2. As the goal of this paper is to understand how GNN predictions respond to the evolution of graphs. However, in this paper, the GNNs are all static GNNs. What is the effectiveness of the proposed method in explaining the dynamic GNN methods? For example, TGN or TGAT?
3. What is the relationship between the Shapley value through temporal dimension? Does the proposed method capture the influence of evolving nature in dynamic graphs?

**Details Of Ethics Concerns:**

No ethical concerns.

---

> ### Author Response · Authors · 2024-11-20
> **Responses to Reviewer 7mCd**
>
> Thank you for taking the time to review our paper! Kindly note that we have uploaded a **revised version**, which includes additional details for your review.
>
> > There are typos in line 221. Please carefully check them.
>
> Thank you for pointing out the typos in line 221. We have carefully reviewed and corrected them in the revised manuscript.
>
> > As the goal of this paper is to understand how GNN predictions respond to the evolution of graphs. However, in this paper, the GNNs are all static GNNs. What is the effectiveness of the proposed method in explaining the dynamic GNN methods? For example, TGN or TGAT?
>
> Our methods focus on the explanations of  static GNNs on evolving graphs, thus they are not directly applicable to dynamic GNNs, such as TGN or TGAT. The main challenge is that the TGN or TGAT use LSTM or GRU model for memory updater. For LSTM or GRU model, we cannot attribute the changes of the ouput to the changes of multiple input. For example, in the GRU model, given an initial graph where the input is $x_1$ and $h_0$, the output is $y_0$. For the end graph with inputs $x_1+\Delta x$ and $h_0+ \Delta h$, the output becomes $y_1$. The change $\Delta y=y_1-y_0$ cannot be attributed to $\Delta x$ and $\Delta h$ using existing rules. The reason is that the rule designed for non-linear function can only attribute $\Delta y$ to $\Delta x$. But they can not attribute the $\Delta y$ to $\Delta x$ and $\Delta h$. Developing rules to attribute output changes to multiple input changes for the nonlinear function is a key direction for future work. After that we can apply our method to dynamic graph models.
>
> Additionally, the problem we define is to explain why the predicted probability distribution changes in a static GCN model when the edge weights in the graph change continuously. We experimentally verify the effectiveness of our proposed method on node classification, link prediction, and graph classification tasks. We do not focus on explaining dynamic graph models.
>
> > What is the relationship between the Shapley value through temporal dimension? Does the proposed method capture the influence of evolving nature in dynamic graphs?
>
> We have the clarification question: In "What is the relationship between the Shapley value through temporal dimension?", what does "the relationship between the Shapley value through temporal dimension" mean? Our understanding of this question is that you may want to ask how Shapley value captures relevant information through temporal dimension? If my understanding of the issue is incorrect, please feel free to let us know. The following is the response to the question based on the understanding outlined above.
>
> In the context of static GNNs, in our problem settings, the evolving nature of dynamic graphs is the continuously changed edge weights. From the perspective of the computational graph, these changed edges lead to altered layer edges and message flows. Due to these all changed edges (layer edges or message flows), the prediction probability distributions are also changed. To capture this influence, we first derive the changed hidden vector in Equation (6). Then, using the multipliers and chain rule from DeepLIFT, we compute the contributions of the altered message flows. The sum of these contributions equals the change in logits.  In this step, we capture the influence of changed message flows.
>
> Next, we use the Shapley value to attribute these contributions to the layer edges. Based on the contribution of changed message flows, the Shapley value allows us to quantify how each changed layer edge contributes to the overall change in the prediction. This process enables us to capture the temporal influence of evolving graph structures on the model's output. In this step, we  capture the influence of changed layer edges.
>
> Finally, we select the small number of chaned layer edgs to explain the evolution of predicted probability distribution. The results in **Figures 5, 8, and 12** demonstrate that our method outperforms other baselines. Thus, our methods capture the influence of the evolving nature in dynamic graphs .

---

> > ### Comment · Reviewer_7mCd · 2024-11-26
> >
> > Thanks for your response. I have increased the score based on the rebuttal.

---

> > > ### Author Response · Authors · 2024-11-26
> > > **Thanks for your acknowledgement!**
> > >
> > > Thank you for taking the time to review our rebuttal. We deeply thank your reading, understanding, and appreciating of our rebuttal!

---

> ### Author Response · Authors · 2024-11-23
> **Gentle reminder for feedback of our response**
>
> Dear Reviewer 7mCd,
>
> We posted our official responses to your questions in the original review. May we humbly ask for your further feedback about our responses? We will be appreciative if we can have an opportunity to learn about your thinking and answer further questions.
>
> Thanks from the anonymous authors!

---

### Official Review · Reviewer_d3Hv · 2024-11-05

**Soundness:** 2
**Presentation:** 3
**Contribution:** 3
**Rating:** 6
**Confidence:** 4

**Summary:**

The paper presents Axiom Layer Edge, a novel explanation framework designed for dynamic graph learning models, particularly focusing on Graph Neural Networks (GNNs). The framework aims to identify the most influential "layer edges" — connections within the GNN layers that have a significant impact on the model’s predictions. This approach is based on the use of Shapley values to fairly attribute contribution scores to individual layer edges and provides a principled and interpretable explanation for prediction shifts that occur as the graph changes over time. The results on several datasets and graph task show that method outperforms other existing baseline methods.

**Strengths:**

- The work focuses on problem of explainability in evolving/dynamic graphs which is an important task
- Extending concepts of shapley value to many dynamic graphs is also key contribution of the work
- The work aims to focus on the trade-offs between quality of interpretablity and fidelity

**Weaknesses:**

- Applying shapley value and KL divergence to large or highly dynamic graphs can be challenge especially due to combinatorial nature of computation
- No theoretical analysis on computation complexity is provided for evolving graphs
- There are several limitations with Shapley such as the lack of robustness, stability etc [1]. It is unclear what are the limitations of Shapley in this setting.

[1] Kumar, I. Elizabeth, Suresh Venkatasubramanian, Carlos Scheidegger, and Sorelle Friedler. "Problems with Shapley-value-based explanations as feature importance measures." In International conference on machine learning, pp. 5491-5500. PMLR, 2020.

**Questions:**

- It would be helpful if the authors can justify the use of Shapley value instead of other game-theoretic methods  (e.g., core, Banzhaf etc)?
- How can we approximate the calculations of shapley and KL divergence in large/dynamic graphs?

- Can we have a discussion on the usefulness of the explanations from the point of view domain experts? Also, are the explanations stable to graph operations like attention, pooling, etc.?

---

> ### Author Response · Authors · 2024-11-20
> **Responses to Reviewer d3Hv**
>
> Thank you for taking the time to review our paper! Kindly note that we have uploaded a **revised version**, which includes additional details for your review.
>
> > Applying shapley value and KL divergence to large or highly dynamic graphs can be challenge especially due to combinatorial nature of computation.
>
> Minimizing the KL-divergence in large dynamic graphs is the challenge, while applying Shapley value in large dynamic graphs may be not the challenge. Because the total number of players in Shapley value corresponds to T, the number of layers in the GNN, which is typically small (e.g., T=2 or T=3 in most GNN applications).  In our experiments, T=2, and $2^T $- 1 is relatively small.
>
> In **Appendix A.7.7**, we present the running time results for the Pubmed, Coauthor-Computer, and Coauthor-Physics large graph datasets in **Figure 9**. As shown, the running time of using the Shapley value to attribute contributions to layer edges is relatively small, even for larger graphs. This is due to two factors: 1) the Shapley value is computed in a vectorized form, which reduces computation time, and 2) some results from the message flow contribution calculation can be reused when computing the Shapley value. **Section 3.4** provides a complexity analysis, which reveals that the complexity of obtaining the changed message flows and selecting important layer edges grows exponentially with the number of changed edges or layer edges. Therefore, selecting the small number of important layer edges to minminze the KLdivergence in the large dynamic graph datasets is the challenge. Besides, efficiently obtaining the changed message flows is also a challenge. Future work may involve designing incremental message flow searches for different graph topologies and using more specific optimization algorithms to accelerate the process.
>
> > No theoretical analysis on computation complexity is provided for evolving graphs.
>
>  We provide the complexity analysis in **Section 3.4**. The complexity of obtaining of the changed message flows is $O\big(|\Delta \mathcal{E}|^T \big)$, where $|\Delta \mathcal{E}|$ is the number of changed edges. The complexity of calculating contributions of message flows and applying Shapley value are $O\big(|\Delta \mathcal{F}| d^1 \cdots d^{t} d^{t+1} \cdots d^{T+1} \big)$ and $O\big(|\Delta \mathcal{F}| (2^T-1) d^1 \cdots d^{t} d^{t+1} \cdots d^{T+1} \big)$, respectively, where  $d^t$ and  $d^{t+1}$ denote the dimension of the $\theta^{t} \in \mathbb{R}^{d^{t} \times d^{t+1}}, t=\{1,\cdots, T\}$. $|\Delta \mathcal{F}|$ denotes the number of changed message flows. The complexity of selecting the inportant layer edges is $O\big(|\Delta \mathcal{A}|^3 \big)$, where $|\Delta \mathcal{A}|$ is the number of changed layer edges.
>
> > There are several limitations with Shapley such as the lack of robustness, stability etc [1]. It is unclear what are the limitations of Shapley in this setting.
>
> As noted in reference [1], using the Shapley value in our experimental setting may lead to out-of-distribution issues. In our method, given the message flow F and coalition S, we obtain a message flow F′ with different layer edge weights. If the GNN model has never encountered an example like F′ during training, its predictions involved F′ will not be relevant to explaining an in-distribution sample, thus affecting the explanations. Even though Shapley value has out-of-distribution issues, in **Figures 5, 8, and 12**, the explanations we obtained using Shapley value can better explains the evolution of the predicted probability distribution than other baselines. Additionally, in Appendix **A.7.8**, we generate motif datasets to verify the accuracy of our explanations, with results shown in **Table 7, Figure 10, and Figure 11**, demonstrating the interpretability of our method.
>
> **Reference**
>
> [1] Kumar, I. Elizabeth, Suresh Venkatasubramanian, Carlos Scheidegger, and Sorelle Friedler. "Problems with Shapley-value-based explanations as feature importance measures." In International conference on machine learning, pp. 5491-5500. PMLR, 2020.

---

> ### Author Response · Authors · 2024-11-20
> **Responses to Reviewer d3Hv**
>
> > It would be helpful if the authors can justify the use of Shapley value instead of other game-theoretic methods (e.g., core, Banzhaf etc)?
>
> We select the Shapley value due to its efficiency. When calculating the contribution of message flows, our method ensures that $\Delta \boldsymbol{z}= \sum_{s=1}^{|\Delta \mathcal{F}|} C_{s}$ . We use the Shapley value to attribute the $C_{s}$ to the contribution of layer edges. Note that the reward of all player sets is $C_{s}$. The contribution of layer edge (each player) is denoted as $\Phi_l$. Using the efficiency property, $\sum_l \Phi_l = C_s$ . Thus, the sum contributions of changed layer edges is equal to the change in logits for the target node/link/graph, i.e, $\sum_{l=1}^{|\Delta \mathcal{A}|}\Phi_{l}=\Delta z$.  This equation is called summation-to-delta property. With this property, we can derive the KL divergence and formulate an optimization problem to select the important layer edges as explanations, highlighting the importance of efficiency.
>
> For the core, in our method, only efficiency is required; Coalitional Rationality is not necessary. For example, considering the three isolated nodes, A,B,C, S={AB} denotes there is edge connect A and B, then the graph structure now is A--B, C. Each S can represent the one graph structure. Give this graph structure, the value function measures the contributions of edges in coalition S to the prediction of node C. Because the edge AB can not pass messages to node C, thus the v(AB)=0. Similarly, S={BC}, the graph structure is B--C, A, supposing the v(BC)=4. If S={AB,BC}, the graph structure is A--B--C. Compared with the graph structure B--C, A, the messages of node A can also affect the node C classification. If A and B together pass the useful message for node C classification, the v(AB,BC)>v(BC), then the v(AB,BC)>v(BC)+v(AB). In this case, it violates the Coalitional Rationality, but this allocation is reasonable. Thus, if the player denotes the  layer edge in the graph, due to connectivity issues, it is not necessary to satisfy Coalitional Rationality.
>
> The Banzhaf does not satisfy efficiency and therefore cannot be used in our method.
>
> > How can we approximate the calculations of shapley and KL divergence in large/dynamic graphs?
>
> As discussed before, in large graphs, the running time for attributing contributions to layer edges using the Shapley value is relatively small. However, with a large number of layers (e.g., T=10), we can random sampling of the coalition S to compute the approximate Shapley value. The cost of selecting important layer edges increases as the number of changed edges grows. Therefore, selecting only a small number of layer edges to minimize KL-divergence presents a challenge. Potential solutions include selecting the top-k layer edges for explanation to approximate the KL-divergence or using a commercial solver to accelerate the process. Additionally, as shown in **Figure 9**, for large graph datasets, when the number of changed layer edges exceeds 20,000, the running time is approximately 400 seconds, which is acceptable.
>
> > Can we have a discussion on the usefulness of the explanations from the point of view domain experts?
>
> For the usefulness of explanations, for example, considering the rumor detection task: in $G_0$, the news is classified as non-rumor, while in $G_1$, as the number of comments increases, it is classified as a rumor. Our explanation algorithm identifies key comments that explain this classification change, enabling managers to understand the model's decisions and address rumors accordingly. In the medical domain, by tracking dynamic health data, [2] can predict a patient's risk of developing a disease. For example, at $t_0$, the model predicts no disease, but at $t_1$, it predicts the patient has the disease. Interpretable algorithms can highlight key medical indicators that explain this prediction change, allowing doctors to make informed decisions and increase their trust in the model.
>
> > Also, are the explanations stable to graph operations like pooling?
>
> The explanations are stable and effective for pooling operations. In the graph classification task, we use average pooling. **Figures 5 and 8** (third row) show the results of this task, where our method outperforms all other baselines.
>
> **Reference**
>
> [2] Placido D, Yuan B, Hjaltelin J X, et al. A deep learning algorithm to predict risk of pancreatic cancer from disease trajectories[J]. Nature medicine, 2023, 29(5): 1113-1122.

---

> ### Author Response · Authors · 2024-11-20
> **Responses to Reviewer d3Hv**
>
> > Also, are the explanations stable to graph operations like attention?
>
> When our method is extended to the GAT model, we can only attribute changes in logits to changes in the attention weights of edges. Then we can select the small number of changed attention weights of edges to explain the evolution of prediction probability distribution. It is similar to attributing changes in prediction probability distribution to changes in layer edges in GCN models. Some work regard the attention weight as the explanations[3][4]. Thus the changed attention weights can also be considered as the explanation to explain the evolution of prediction probability distribution.
>
> But we cannot explain which added or removed edges contribute more or less to the changed attention weights, and how these added or removed edges impacts the evolution of prediction probability distribution. The reason is that we do not define the rules to attribute the difference of softmax function to the changed input. For example, we consider a target node J with four neighbors I,K,L,M. The attention for edge J--I is $\frac{e^a}{e^a+e^b+e^c+e^d}$, if we remove the edge J--L and the edge J--M, the attention for the IJ now is $\frac{e^a}{e^a+e^b}$. The problem is that we do not have the rules to attribute the $\frac{e^a}{e^a+e^b+e^c+e^d}-\frac{e^a}{e^a+e^b}$ to the changed input c and d. Thus, designing this rule is our future work. After obtaining this rule, we can explain the GAT model.
>
> **Reference**
>
> [3] Danish Pruthi, Mansi Gupta, Bhuwan Dhingra, Graham Neubig, and Zachary C. Lipton. 2020. Learning to deceive with attention-based explanations. In Proceedings of the 58th Annual Meeting of the Association for Computational Linguistics, pages 4782– 4793, Online. Association for Computational Linguistics.
>
> [4]Shikhar Vashishth, Shyam Upadhyay, Gaurav Singh Tomar, and Manaal Faruqui. 2019. Attention interpretability across NLP tasks. arXiv preprint arXiv:1909.11218.

---

> ### Comment · Reviewer_d3Hv · 2024-11-20
> **Thanks!**
>
> Dear Authors,
>
> Thanks for your effort in the rebuttal. I have increased my score based on the rebuttal and the scope of the paper.
> One quick clarification question: do you really need the efficiency property (regarding Banzhaf)?

---

> > ### Author Response · Authors · 2024-11-21
> > **Thanks for your acknowledgement!**
> >
> > For the question: do you really need the efficiency property (regarding Banzhaf)?
> >
> > Answer:  Yes, we really need the efficiency property. Due to the efficiency property, we can make sure the sum contirbutions of all changed layer edges is equal to the changed logits, i.e. $\sum_{l=1}^{|\Delta \mathcal{A}|}\Phi_{l}=\Delta z$ . Based on this equation, we can derive the KL divergence and select the important layer edges as explanations. Therefore, the efficiency property is crucial to  our method.
> >
> > We deeply thank your reading, understanding, and appreciating of our rebuttal!

---

### Meta-Review · Area_Chair_aNEV · 2024-12-17

**Metareview:**

This paper presents a novel method for explaining predictions in evolving graphs, focusing on Graph Neural Networks (GNNs) where edge weights change over time. The approach uses Shapley values to attribute contributions to key layer edges and provides interpretable explanations. Experimental results demonstrate its superior performance in tasks like node classification, link prediction, and graph classification.

Reviewers praised the innovation but raised concerns about computational complexity, application to dynamic GNNs, and the robustness of explanations. The authors' responses addressed most of these concerns, and the paper is recommended for acceptance.

**Additional Comments On Reviewer Discussion:**

Reviewers praised the innovation but raised concerns about computational complexity, application to dynamic GNNs, and the robustness of explanations. The authors' responses addressed most of these concerns.

---

### Decision · Program_Chairs · 2025-01-22

Accept (Poster)